# CDQUANT: ACCURATE POST-TRAINING WEIGHT QUANTIZATION OF LLMS USING GREEDY COORDINATE DESCENT

## ABSTRACT

Large language models (LLMs) have recently demonstrated remarkable performance across diverse language tasks. But their deployment is often constrained by their substantial computational and storage requirements. Quantization has emerged as a key technique for addressing this challenge, enabling the compression of large models with minimal impact on performance. The recent GPTQ algorithm, a post-training quantization (PTQ) method, has proven highly effective for compressing LLMs, sparking a wave of research that leverages GPTQ as a core component. Recognizing the pivotal role of GPTQ in the PTQ landscape, we introduce CDQuant, a simple and scalable alternative to GPTQ with improved performance. CDQuant uses greedy coordinate descent to minimize the layer-wise reconstruction loss to achieve high-quality quantized weights. Our algorithm is easy to implement and scales efficiently to models with hundreds of billions of parameters. We perform extensive evaluation on Gemma, and PaLM2 model families, and demonstrate that CDQuant consistently outperforms GPTQ in 2-4 bit weight quantization. Moreover, CDQuant improves the performance of state-of-the-art PTQ techniques such as QuIP and FrameQuant when used as a replacement for their GPTQ component, resulting in further gains in quality.

## 1 INTRODUCTION

Large language models (LLMs) have shown remarkable ability to handle various language tasks (Touvron et al., 2023; OpenAI, 2023; Google, 2023), but their widespread adoption is hampered by their substantial computational and memory demands. To tackle this challenge, researchers have explored techniques like quantization, pruning, and distillation, with quantization being a particularly promising avenue for reducing model size and inference time without significantly sacrificing performance (Miao et al., 2023). Quantization techniques broadly fall into two categories: post-training quantization (PTQ) and quantization-aware training (QAT). QAT, while potentially yielding better results, poses a significant challenge for LLMs due to the immense resources required to train these large models. As a result, a growing body of research has focused on PTQ, which is generally less computationally intensive and can be applied to large, pre-trained models.

A seminal work in PTQ for LLMs is GPTQ (Frantar et al., 2022), which introduced a one-shot weight quantization approach that minimizes the following layer-wise output reconstruction loss: $\|X(W - \hat{W})\|_F^2$, where $X = [\mathbf{x}_1, \mathbf{x}_2 \ldots \mathbf{x}_n]$ represents the layer inputs and $W, \widehat{W} \in \mathbb{R}^{d_{\text{in}} \times d_{\text{out}}}$ are the original and quantized weight matrices, respectively. This method has sparked a wave of research in PTQ. Subsequent works have built upon GPTQ, addressing its limitations and improving it's performance. For instance, SpQR (Dettmers et al., 2023) and OWQ (Lee et al., 2023) improve GPTQ by explicitly addressing outlier weights, maintaining them in full precision, while quantizing the remaining weights using GPTQ. This hybrid approach improves quantization accuracy, particularly at lower bit-widths. Another line of research focuses on transforming the weight space before applying GPTQ. For instance, QuIP (Chee et al., 2024), FrameQuant (Adepu et al., 2024) transform the weights into a more quantization-friendly space and apply GPTQ in the transformed space. AWQ (Lin et al., 2023), SmoothQuant (Xiao et al., 2023) reduce the effect of activation outliers by performing feature scaling before quantizing the weights using standard techniques.

Given the central role GPTQ plays in the landscape of PTQ methods for LLMs, improving its quality directly translates to better quantization across numerous techniques that build upon it. Consequently, in this work, we revisit the core optimization problem addressed by GPTQ and introduce a novel coordinate descent based algorithm, CDQuant, to minimize the objective. CDQuant is an iterative optimization technique that greedily selects coordinates to descend along in each iteration. This approach contrasts with GPTQ, which cycles through the coordinates only *once*, in a predetermined order, often leading to suboptimal solutions for the layer-wise objective (see Section 2 for more details). CDQuant is simple to implement and scales effectively to models with billions of parameters. We further extend CDQuant to group/sub-channel quantization, a technique gaining popularity of late (Dettmers et al., 2021; Dettmers & Zettlemoyer, 2023).

Extensive experiments on Gemma (Mesnard et al., 2024; Gemma-Team, 2024), and PaLM2 (Anil et al., 2023) model families demonstrate that CDQuant consistently outperforms GPTQ across various model sizes and quantization precision levels (2-4 bits). Notably, for INT2 quantization of PaLM2-Otter, CDQuant achieves a $10\%$ perplexity reduction compared to GPTQ. Furthermore, integrating CDQuant with QuIP (Chee et al., 2024) and FrameQuant (Adepu et al., 2024) improves their performance by $\sim 5\%$ for INT2 quantization of the Gemma-2 27B model, showcasing its potential as a drop-in replacement for GPTQ in existing PTQ techniques.

## 2 RELATED WORK

The literature on quantization is huge. In this section, we only review the works that are related to quantization of large pre-trained models, and those that are related to our work.

**GPTQ.** Inspired by the Optimal Brain Surgeon (OBS) framework for pruning (LeCun et al., 1989), Optimal Brain Quantization (OBQ) (Frantar & Alistarh, 2022) emerged as a promising post-training quantization (PTQ) technique. However, OBQ remains computationally expensive and struggles to scale effectively to large language models (LLMs). This is because of a complicated step involved in OBQ which requires updating all the unquantized coordinates in each step (see Equation 2 in Frantar et al. (2022)). Implementing this step in GPUs is extremely slow because of the sheer number of gathers/scatters that need to be performed at each step. Frantar et al. (2022) subsequently introduced GPTQ, which employs heuristics to accelerate the OBQ algorithm. Specifically, GPTQ updates coordinates once in a cyclic manner rather than the greedy approach used in OBQ. This modification significantly speeds up the algorithm but comes at the cost of reduced quality. In our work, we address this performance drop by proposing greedy coordinate descent algorithms that are both straightforward and easy to implement.

**Other Weight-only Quantization Techniques.** Recent works on post-training quantization have sought to improve upon GPTQ. One popular strategy here is to identify and isolate problematic weights before applying GPTQ. For example, SpQR (Dettmers et al., 2023) isolates outlier weights and keeps them at full precision, quantizing only the remaining values using GPTQ. Similarly, OWQ (Lee et al., 2023) identifies weights that are highly sensitive to small perturbations and excludes them from the GPTQ quantization process. JSQ (Guo et al., 2024) clips activations (where the clipping thresholds are searched through simulated annealing), following which a combination of activation range and weight saliency are used to sparsify the weight matrix before quantizing it with GPTQ. SqueezeLLM (Kim et al., 2024) also preserves outliers in floating point, however, employs non-uniform quantization by performing K-means clustering on the final end-to-end loss (where the gradient is approximated with zero,and the Hessain with the Fisher information matrix). Any-Precision LLM (Park et al., 2024) improves SqueezeLLM by allowing several nested precisions through a control over the number of cluster centers. However, both these approaches involve storing a codebook, hence making hardware and software compatibility tricky.

Another promising direction involves transforming the weights into a different basis before quantization. QuIP (Chee et al., 2024), and FrameQuant (Adepu et al., 2024), SpinQuant (Liu et al., 2024), QuaRot (Ashkboos et al., 2024) take this route and perform quantization in the transformed space using GPTQ. In all these approaches, we could potentially substitute GPTQ with CDQuant and expect to see further improvements in performance (see Section 4 for empirical evidence). Other promising methods such as AWQ (Lin et al., 2023), AffineQuant (Ma et al., 2024), suppress the effect of outliers in input activations by adjusting their scale and transferring it to weights. They then perform the standard MinMax quantization on the rescaled weights. One could replace MinMax with

GPTQ, CDQuant to improve the performance of these techniques (see Table 4 for a comparison of CDQuant with AWQ).

While working on our paper, we came across QuantEase (Behdin et al., 2023), a parallel research effort sharing similar goals as ours for improving GPTQ. While both methods leverage the concept of coordinate descent, QuantEase adopts a cyclic approach for weight updates, whereas our work employs a greedy strategy. Although our experiments (Appendix D) indicate comparable performance between the two methods, our work extends beyond QuantEase by introducing specialized algorithms for group/sub-channel quantization, not just full-channel quantization. Additionally, we develop novel block coordinate descent algorithms that further improve the performance. That being said, both these works (QuantEase and ours) collectively highlight the potential of coordinate descent algorithms in outperforming GPTQ.

**Vector Quantization.** AQLM (Egiazarian et al., 2024) generalizes Additive Quantization to LLMs by introducing input-adaptiveness (through calibration data) and jointly optimizing the codebooks across transformer blocks. QuIP# (Tseng et al., 2024) improves QuIP's incoherence processing through random Hadamard matrices, introduces vector quantization, and adds an additional layer of finetuning to improve model quality. However, in this work, our focus is methods that improve scalar quantization.

**Weight+Activation Quantization.** SmoothQuant (Xiao et al., 2023), OS+(Wei et al., 2023; 2022) have a similar flavour as AWQ, but quantize both weights and activations after scaling them appropriately. OmniQuant (Shao et al., 2023) performs quantization of the entire transformer block in a single shot. This encompasses both activation and weight quantization. Furthermore, it subsumes SmoothQuant, OS+ by using both feature scaling and outlier suppression. QLLM (Liu et al., 2023) tackles the issue of outliers in the activations by splitting the outlier features into multiple sub-channels and then recombining them, effectively reducing their influence. QLLM also incorporates a fine-tuning step at the end, introducing low-rank weights into each layer of the LLM. LLM.int8() (Dettmers et al., 2022) quantizes both acativations and weights to 8-bits and also identifies outliers and stores them in full precision. LQER (Zhang et al., 2024) approximates the weight quantization error with the product of two low rank matrices.

**Efficient End-to-End Quantization.** To recover the drop in performance from quantization Chai et al. (2023); Dettmers et al. (2024); Hu et al. (2021) perform low-rank parameter-efficient fine-tuning.

## 3 CDQUANT

**Notation.** Throughout the paper, we denote vectors by bold faced letters ($\mathbf{a}$), and matrices by capital letters ($A$). $\|\mathbf{a}\|_2 = \sqrt{\sum_i \mathbf{a}_i^2}$ is the Euclidean norm and $\|A\|_F = \sqrt{\sum_{i,j} A_{ij}^2}$ is the Frobenius norm of a matrix. $\mathrm{diag}(\mathbf{a})$ represents a diagonal matrix with $\mathbf{a}$ as its diagonal entries. $d_{\text{in}}, d_{\text{out}}$ denote the input, output dimensions of a layer. $W \in \mathbb{R}^{d_{\text{in}} \times d_{\text{out}}}$ is the weight matrix of the layer, and $X = [\mathbf{x}_1, \mathbf{x}_2 \dots \mathbf{x}_n] \in \mathbb{R}^{n \times d_{\text{in}}}$ is the matrix containing $n$ datapoints that are sent as input to the layer. $H = X^T X$ is the Hessian matrix for our objective in Equation (1). $c$ denotes the number of bits of precision used in quantization. In quantization, we aim to represent $W$ as $Q \times \mathrm{diag}(\mathbf{a}) + \mathbf{1b}^T$, where $\mathbf{a}, \mathbf{b} \in \mathbb{R}^{d_{\text{out}}}$ represent the scale and bias parameters and $Q \in \{0, 1, \dots 2^c - 1\}^{d_{\text{in}} \times d_{\text{out}}}$ is the quantized matrix.

Many existing PTQ techniques, including GPTQ, aim to solve the following layer-wise optimization objective: $\min_{\mathbf{a},\mathbf{b},Q} \|X(W - Q \times \mathrm{diag}(\mathbf{a}) - \mathbf{1b}^T)\|_F^2$. Observe that this problem breaks down into $d_{\text{out}}$ independent problems across the output dimension. So, in the sequel, we focus on the following problem of quantizing a $d_{\text{in}}$-dimensional vector

$$\min_{a,b,\mathbf{q}} \|X(\mathbf{w} - a\mathbf{q} - b)\|_2^2. \tag{1}$$

For a fixed $(a, b)$, this problem is called *Integer Linear Regression* problem. It turns out, finding optimal solutions to this problem is NP-hard (Chrétien & Corset, 2009; Park & Boyd, 2018). So, several works have designed heuristics to solve this problem (Nagel et al., 2020; Li et al., 2021; Frantar & Alistarh, 2022; Frantar et al., 2022; Hubara et al., 2021). Within the context of LLMs, GPTQ is perhaps the most popular among these techniques, as it scales efficiently to models with billions of parameters (Frantar et al., 2022). In this work, we aim to improve upon GPTQ by designing better

---

**Algorithm 1** Greedy Coordinate Descent (CD)

---

1: **Input:** $T$ - coordinate descent steps, $X$ - input data matrix, $\mathbf{w}$ - vector to be quantized, $a$ - scale, $b$ - bias, $\mathbf{q}_0$ - initial estimate
2: Compute Hessian $H$ as: $H \leftarrow X^T X$
3: Compute gradient $\mathbf{g}$ at $\mathbf{q}_0$ as: $\mathbf{g} \leftarrow 2H(\mathbf{q}_0 - a^{-1}(\mathbf{w} - b))$
4: **for** $t \in [1 : T]$ **do**
5:    Find the coordinate that leads to the largest reduction in loss

$$i^*, r^* = \underset{i \in \{0,1,\dots d_{\text{in}}-1\}, r \in \{0,1,\dots 2^c-1\}}{\arg\min} (r - q_{t-1,i})^2 H_{i,i} + (r - q_{t-1,i}) \mathbf{g}_i$$

6:    Update gradient $\mathbf{g}$ as

$$\mathbf{g} \leftarrow \mathbf{g} + 2(r^* - q_{t-1,i^*}) H_{i^*,\cdot},$$

where $H_{i^*,\cdot}$ is the $i^*$ column of $H$
7:    Update $\mathbf{q}_{t-1}$ as

$$\mathbf{q}_t \leftarrow \mathbf{q}_{t-1} + (r^* - q_{t-1,i^*}) \mathbf{e}_{i^*},$$

where $\mathbf{e}_{i^*}$ is the standard basis vector with 1 in $i^*$ position and 0 everywhere else
8: **end for**

---

heuristics, while maintaining its scalability aspect. To this end, we rely on performing coordinate descent on objective (1), which we describe below.

### 3.1 GREEDY COORDINATE DESCENT

In this section, we assume we have suitable values for scale ($a$) and bias ($b$) parameters already at hand, and focus on optimizing $\mathbf{q}$. For a more in-depth discussion of how we determine these values, please refer to the final part of the section. As the name suggests, in greedy coordinate descent, at each round, we find the coordinate that leads to the biggest reduction in the objective and descend along that coordinate. Letting $\mathcal{L}(\mathbf{q}) := \|X(\mathbf{w} - a\mathbf{q} - b)\|_2^2$ be the objective in Equation (1), we try to find a coordinate $i$ and value $r$, such that updating the $i^{th}$ coordinate to $r$ gives the biggest reduction in loss

$$\min_{i,r} \mathcal{L}(\mathbf{q} + (r - q_i)\mathbf{e}_i) - \mathcal{L}(\mathbf{q}),$$

where $q_i$ is the $i^{th}$ element of $\mathbf{q}$, and $\mathbf{e}_i$ is the standard basis vector with 1 in $i$ position and 0 everywhere else. Luckily for us, this can be implemented extremely efficiently as we have analytical expressions for the objective. In particular, one can easily show that

$$\mathcal{L}(\mathbf{q} + (r - q_i)\mathbf{e}_i) - \mathcal{L}(\mathbf{q}) = (r - q_i)^2 H_{i,i} + (r - q_i) \mathbf{g}_i,$$

where $H, \mathbf{g}$ are the Hessian and gradient of $\mathcal{L}$ evaluated at $\mathbf{q}$. This follows from the fact that $\mathcal{L}$ is a quadratic function in $\mathbf{q}$. Algorithm 1 describes this procedure (note that line 5 is the key step which identifies the descent direction).

**Remark 1** (Comparison with OBQ). *Similar to our technique, OBQ also updates its coodinates in a greedy manner. But it differs from our algorithm in one crucial step: it updates all the unquantized coordinates in each step (see Equation 2 in Frantar et al. (2022)). Implementing this step on accelerators such as GPUs/TPUs is extremely slow because of the sheer number of gathers/scatters that need to be performed at each step. In contrast, our algorithm only requires updating one coordinate at a time.*

**Extension to Block Coordinate Descent.** A natural extension to Algorithm 1 is block coordinate descent (BCD), where multiple coordinates are updated simultaneously in each iteration. While a greedy approach to BCD could, in principle, optimize the objective much better, the computational cost becomes prohibitive. Specifically, updating $k$ coordinates at a time necessitates evaluating $(d_{\text{in}} \times 2^c)^k$ possible combinations of coordinates and their corresponding values. To address this, we propose a randomized BCD strategy (see Algorithm 2), which partitions the coordinates into random $d_{\text{in}}/k$ blocks and searches only over these blocks. This significantly reduces the search space to a more manageable $d_{\text{in}}/k \times 2^{kc}$ possibilities, making the algorithm practical for larger models. In our experiments, we primarily use $k = 2$ and $c \in \{2, 3, 4\}$. Note that Algorithm 2 with $k = 1$ recovers Algorithm 1.

---

**Algorithm 2** Block Coordinate Descent with Random Blocks (BCD)

---

1: **Input:** $T$ - coordinate descent steps, $k$ - block size, $X$ - input data matrix, $\mathbf{w}$ - vector to be quantized, $a$ - scale, $b$ - bias, $\mathbf{q}_0$ - initial estimate

2: Compute Hessian $H$ as: $H \leftarrow X^T X$

3: Compute gradient $\mathbf{g}$ as: $\mathbf{g} \leftarrow 2H(\mathbf{q}_0 - a^{-1}(\mathbf{w} - b))$

4: **for** $t \in [1:T]$ **do**

5:     Randomly partition the set $\{0, 1, \dots d_{\text{in}} - 1\}$ into $d_{\text{in}}/k$ blocks, each of size $k$

6:     Find the block that leads to the largest reduction in loss

$$i^*, \mathbf{r}^* = \operatorname*{arg\,min}_{\substack{i \in \{0,1,\dots d_{\text{in}}/k-1\}, \\ \mathbf{r} \in \{0,1,\dots 2^c-1\}^k}} (\mathbf{r} - \mathbf{q}_{t-1,i})^T H_{i,i} (\mathbf{r} - \mathbf{q}_{t-1,i}) + (\mathbf{r} - \mathbf{q}_{t-1,i})^T \mathbf{g}_i,$$

    where $\mathbf{q}_{t-1,i}, H_{i,i}$ are the sub-vector, sub-matrix of $\mathbf{q}_{t-1}, H$ corresponding to block $i$.

7:     Update gradient $\mathbf{g}$ as

$$\mathbf{g} \leftarrow \mathbf{g} + 2H_{i^*,\cdot}(\mathbf{r}^* - \mathbf{q}_{t-1,i^*}),$$

8:     Update $\mathbf{q}_{t-1}$ as

$$\mathbf{q}_{t-1,i^*} \leftarrow \mathbf{r}^*, \quad \mathbf{q}_t \leftarrow \mathbf{q}_{t-1},$$

9: **end for**

---

**Initializing** $a, b, \mathbf{q}$. To initialize $a, b, \mathbf{q}$ in Algorithms 1, 2, we introduce a technique called Optimal Weight Clipping (OWC), which draws inspiration from the Learnable Weight Clipping (LWC) mechanism used in OmniQuant (Shao et al., 2023). In OWC, we quantize weight $\mathbf{w}$ as follows

$$\mathbf{q} = \text{clamp}\left(\left\lfloor \frac{\mathbf{w} - b}{a} \right\rceil, 0, 2^c - 1\right), \quad a = \frac{\gamma(\max(\mathbf{w}) - \min(\mathbf{w}))}{2^c - 1}, \quad b = \min(\mathbf{w}). \quad (2)$$

Here, $\gamma \in [0, 1]$ represents the clipping strength. We determine the optimal $\gamma$ by minimizing the following layer-wise loss:

$$\min_{\gamma \in [0,1]} \|X(\mathbf{w} - a\mathbf{q} - b)\|_2^2.$$

Note that while not explicitly stated, both $\mathbf{q}, a$ in the above objective are implicitly dependent on $\gamma$. This optimization can be efficiently solved using a simple grid search. In contrast, LWC (Shao et al., 2023) optimizes a different objective function, focusing on end-to-end quantization of entire transformer block using gradient-based techniques. It is worth noting that setting $\gamma = 1$ in OWC recovers the widely used MinMax quantization scheme, which is used in many existing quantization methods, including GPTQ (Frantar et al., 2022), SmoothQuant(Xiao et al., 2023). However, MinMax quantization is susceptible to outlier weights, and a smaller $\gamma$ often yields superior results. In our experiments, we observed that OWC provides a much better initialization compared to MinMax quantization, for both GPTQ and CDQuant, suggesting its broader applicability.

### 3.2 EXTENSION TO SUB-CHANNEL QUANTIZATION

In this section, we consider sub-channel (or group) quantization, which is a more fine-grained quantization that divides the weight vector $\mathbf{w}$ into multiple groups and assigns a quantization scale to each group (Dettmers et al., 2021; Dettmers & Zettlemoyer, 2023). Letting $g$ be the group size, we divide weight $\mathbf{w}$ into $d_{\text{in}}/g$ groups $\{\mathbf{w}^{(0)}, \mathbf{w}^{(1)}, \dots \mathbf{w}^{(d_{\text{in}}/g-1)}\}$ each of size $g$, and quantize $\mathbf{w}^{(i)}$ as $a^{(i)}\mathbf{q}^{(i)} + b^{(i)}$. To learn the optimal parameters for this sub-channel quantization, we solve the following optimization problem:

$$\min_{\{a^{(i)}, b^{(i)}, \mathbf{q}^{(i)}\}_{i=0}^{d_{\text{in}}/g-1}} \left\| \sum_i X^{(i)}(\mathbf{w}^{(i)} - a^{(i)}\mathbf{q}^{(i)} - b^{(i)}) \right\|_2^2. \quad (3)$$

Here, $X^{(i)}$ represents the columns of $X$ corresponding to the indices within group $i$. To solve this optimization problem, we employ a coordinate descent approach, similar to Algorithms 1 and 2 described earlier. That is, given initial values for the scaling and bias parameters, we iteratively optimize the quantized representation $\mathbf{q}$ using coordinate descent. Due to space constraints, we present the resulting algorithms in Appendix E (see Algorithms 4, 5).

---

**Algorithm 3** Coordinate Descent for Optimal Weight Clipping (OWC-CD)

---

1: **Input:** $T$ - coordinate descent steps, $g$ - group size, $X$ - input data matrix, $\mathbf{w}$ - weight vector, $\Gamma$-grid of possible values for clipping strength
2: **for** $\beta \in \Gamma$ **do**
3:     **for** $i \in [0 : d_{\text{in}}/g - 1]$ **do**
4:         Compute quantization *residual* $\Delta(i, \beta)$ for group $i$ with clipping strength $\beta$ as:

$$\Delta(i, \beta) \leftarrow \mathbf{w}^{(i)} - a^{(i)}(\beta)\mathbf{q}^{(i)}(\beta) - b^{(i)}(\beta),$$

        where $a^{(i)}(\beta), b^{(i)}(\beta), \mathbf{q}^{(i)}(\beta)$ are as defined in Equation (2).
5:     **end for**
6: **end for**
7: Initialize clipping strengths for each group $\gamma^{(0)}, \ldots \gamma^{(d_{\text{in}}/g - 1)}$
8: Compute Hessian $H$ as: $H \leftarrow X^T X$
9: $\mathbf{v}_i \leftarrow -2H_i(a\mathbf{q} + b - w)$     where $H_i$ is the sub-matrix of $H$ corresponding to the columns of group $i$.
10: **for** $t \in [1 : T]$ **do**
11:     Find the group that leads to the largest reduction in loss

$$i^*, \beta^* = \underset{\substack{i \in \{0,1,\ldots d_{\text{in}}/g-1\}, \\ \beta \in \Gamma}}{\arg\min} (\Delta(i, \gamma^{(i)}) - \Delta(i, \beta))^T H_{i,i}(\Delta(i, \gamma^{(i)}) - \Delta(i, \beta))$$

$$+ \mathbf{v}_i^T(\Delta(i, \gamma^{(i)}) - \Delta(i, \beta))$$

    where $H_{i,i}$ is the block diagonal element of $H$ corresponding to group $i$.
12:     Update $\mathbf{v} \leftarrow \mathbf{v} + 2(\Delta(i^*, \gamma^{(i^*)}) - \Delta(i^*, \beta^*))^T H_{i^*}$
13:     Update $\gamma^{(i^*)} \leftarrow \beta^*$
14: **end for**

---

**Initialization.** Next, we tackle the initialization of parameters $\{a^{(i)}, b^{(i)}, \mathbf{q}^{(i)}\}_{i=0}^{d_{\text{in}}/g-1}$ for our coordinate descent procedure. Our approach draws inspiration from the OWC algorithm described above, adapting its core idea to this problem. In essence, we reframe the initialization problem as one of selecting optimal clipping strengths $\{\gamma^{(i)}\}_{i=0}^{d_{\text{in}}/g-1}$ for groups $\{0, \ldots d_{\text{in}}/g - 1\}$. This leads us to the following problem

$$\min_{\gamma^{(0)}, \ldots \gamma^{(d_{\text{in}}/g-1)}} \Big|\Big| \sum_{i=0} X^{(i)}(\mathbf{w}^{(i)} - a^{(i)}\mathbf{q}^{(i)} - b^{(i)}) \Big|\Big|_2^2, \tag{4}$$

where

$$\mathbf{q}^{(i)} = \text{clamp}\left(\left\lfloor \frac{\mathbf{w}^{(i)} - b^{(i)}}{a^{(i)}} \right\rceil, 0, 2^c - 1\right), \ a^{(i)} = \frac{\gamma^{(i)}(\max(\mathbf{w}^{(i)}) - \min(\mathbf{w}^{(i)}))}{2^c - 1}, \ b^{(i)} = \min(\mathbf{w}^{(i)}).$$

While not explicitly stated, both $\mathbf{q}^{(i)}, a^{(i)}$ implicitly depend on the clipping strength $\gamma^{(i)}$. We use greedy coordinate descent to optimize Equation (4). In each iteration, we update the $\gamma^{(i)}$ that leads to biggest drop in loss (lines 10-14 of Algorithm 3). This procedure, which we call OWC-CD, is described in Algorithm 3.

## 4 EXPERIMENTS

In this section, we first establish that CDQuant achieves superior quantization quality compared to GPTQ. Subsequently, we show its versatility by successfully integrating it as a drop-in replacement for GPTQ within existing PTQ methods such as AWQ (Lin et al., 2023), QuIP (Chee et al., 2024), FrameQuant (Adepu et al., 2024).

### 4.1 COMPARISON WITH GPTQ

We first present the experimental setup and then move on to our results. In all the experiments in this section, the attention layers are quantized to INT8 using MinMax quantization, and low-

Table 1: Table presents the perplexity evaluations for GPTQ, CD, BCD for INT2 quantization of FFN weights.

| Method | Epochs | Gemma-1 7B | Gemma-2 9B | Gemma-2 27B | PaLM2-Otter | PaLM2-Bison |
|---|---|---|---|---|---|---|
| w16a16 | | 10.384 | 10.683 | 8.682 | 5.984 | 5.298 |
| w2a16g128 | | | | | | |
| GPTQ | - | 375.153 | 13.785 | 12.181 | 10.816 | 7.230 |
| CD | 1 | 75.55 | 13.709 | 11.966 | 9.917 | 7.123 |
| BCD(k=2) | 1 | **68.732** | **13.662** | **11.873** | **9.822** | **7.094** |

bit quantization schemes are only applied to the feed-forward layers (FFN). See Appendix F for experiments with quantized attention and FFN layers.

**Models.** Our experiments leverage two families of language models: the open-source Gemma models (Mesnard et al., 2024) and the proprietary PaLM 2 models (Anil et al., 2023). Specifically, we use Gemma-1 7B, Gemma-2 9B, and Gemma-2 27B from the Gemma family, and PaLM2-Gecko, PaLM2-Otter, and PaLM2-Bison from the PaLM2 family. Within the PaLM 2 family, the models increase in size from Gecko to Otter to Bison.

**Baselines.** Since our primary goal is to demonstrate that CDQuant gets improved performance over GPTQ, we have chosen it as the primary baseline in most of our experiments in this section. For all our experiments, we initialize GPTQ with OWC, and run GPTQ for $T = d_{in}$ steps. To ensure stability and generalization, GPTQ regularizes its Hessian matrix by adding a scaled identity matrix ($\lambda I$). Tuning this $\lambda$ for every (model, layer) pair is infeasible. So, we determine a single optimal value using the PaLM2-Gecko model, and apply it universally.

**CDQuant.** We evaluate both the coordinate descent variants described in Section 3.1: CD (Algorithm 1), BCD (Algorithm 2). For per-channel quantization, we initialize CD with Optimal Weight Clipping (OWC), and for sub-channel quantization, we additionally include initialization with OWC-CD. BCD is always initialized with CD in our experiments. Unless otherwise stated, both CD and BCD are run for $T = d_{in}$ iterations, OWC-CD is run for $d_{in}/g$ iterations, where $g$ is the group size. Similar to GPTQ, we regularize the Hessian matrix used in CD, BCD by adding $\lambda I$. We determine a reasonable value for $\lambda$ using the PaLM2-Gecko model, and use it in all our experiments.

**Evaluation.** Following recent works (Frantar et al., 2022; Ma et al., 2024), we evaluate all algorithms using two key metrics: perplexity and downstream task performance. For Gemma models, following Frantar et al. (2022), we calculate perplexity on C4's (Raffel et al., 2019) validation set. For PaLM2 models, we calculate perplexity using a 100 million token subset derived from the PaLM2 training mixture. We use TriviaQA (Joshi et al., 2017), SQuAD (Rajpurkar et al., 2018), NaturalQuestions (Kwiatkowski et al., 2019) and WebQuestions (Berant et al., 2013) to evaluate *generation* capabilities of the quantized models, and to evaluate their *reasoning* capabilities, we test on ARC-c, ARC-e (Clark et al., 2018), HellaSwag (Zellers et al., 2019), BoolQ (Clark et al., 2019), PIQA (Bisk et al., 2020) and WinoGrande (Sakaguchi et al., 2020). For downstream evaluations, we consider the *zero-shot* setting. Finally, to determine which optimization technique is most effective at solving the layer-wise objective in Equation (1), we evaluate the final solution's objective value using the same 100 million tokens that we used to compute perplexity.

**Training.** All techniques are calibrated using 1280 data points, where each data point has 2048 tokens. For OWC, we use a grid size of 50 to find the most optimal $\gamma$. We used 8 Nvidia H100 GPUs for quantizing the models.

**Results for 2 bit quantization.** Table 1 presents the INT2 perplexity numbers for different quantization techniques applied to FFN layers. It can be seen that both our CD and BCD methods have a clear advantage over GPTQ, leading to lower perplexity scores for all models and quantization levels. For example, on PaLM-2 Otter, we see almost $10\%$ improvement in perplexity over GPTQ.

**Results for 3,4 bit quantization.** Table 2 presents the perplexity numbers for different quantization techniques on 3,4 bit quantization. It can be seen that both our CD and BCD methods have a clear advantage over GPTQ, leading to lower perplexity scores for all models and quantization levels. The difference is more pronounced for lower bit quantization.

**Downstream Evals.** Downstream evaluations for the PaLM2 and Gemma model families are presented in Tables 7 and 8, respectively. These tables report the average performance across

Table 2: Table presents the perplexity evaluations for GPTQ, CD, BCD for INT3, INT4 quantization of FFN weights. $wx, ay, gz$ in the config column corresponds to $x$-bit weights, $y$-bit activations and group/sub-channel size of $z$.

| Config | Method | Gemma-1 7B | Gemma-2 9B | Gemma-2 27B | PaLM2 Gecko | PaLM2 Otter | PaLM2 Bison |
|---|---|---|---|---|---|---|---|
| w16a16 | - | 10.348 | 10.683 | 8.682 | 7.948 | 5.984 | 5.298 |
| w3a16 | OWC | 2.885$e$4 | 11.666 | 11.823 | 12.570 | 17.928 | 6.169 |
| | GPTQ | 48.157 | 11.382 | 9.681 | 11.347 | 7.176 | 5.774 |
| | CD | 19.614 | 11.301 | 9.551 | 10.920 | 7.002 | 5.739 |
| | **BCD(k=2)** | **18.552** | **11.284** | **9.526** | **10.898** | **6.979** | **5.733** |
| w3a16g128 | OWC | 21.815 | 11.338 | 9.655 | 11.597 | 8.342 | 5.847 |
| | GPTQ | 15.561 | 11.193 | 9.260 | 10.414 | 6.635 | 5.677 |
| | CD | 13.496 | 11.182 | 9.237 | 10.273 | 6.655 | 5.656 |
| | BCD(k=2) | 13.501 | 11.180 | 9.214 | 10.259 | 6.545 | 5.654 |
| | OWC-CD | 14.827 | 11.220 | 9.290 | 10.706 | 6.635 | 5.686 |
| | OWC-CD + CD | 13.042 | 11.131 | **9.194** | 10.143 | 6.528 | 5.650 |
| | **OWC-CD + BCD(k=2)** | **13.004** | **11.131** | 9.199 | **10.138** | **6.527** | **5.647** |
| w4a16 | OWC | 26.438 | 10.929 | 9.252 | 8.946 | 6.693 | 5.475 |
| | GPTQ | 13.355 | 10.896 | 8.923 | 8.764 | 6.249 | 5.417 |
| | CD | 12.142 | **10.860** | 8.910 | 8.694 | 6.195 | 5.407 |
| | **BCD(k=2)** | **12.048** | 10.863 | **8.899** | **8.691** | **6.192** | **5.405** |
| w4a16g128 | OWC | 11.598 | 10.82 | 8.870 | 8.613 | 6.264 | 5.401 |
| | GPTQ | 11.086 | 10.784 | 8.786 | 8.498 | 6.112 | 5.377 |
| | CD | 10.838 | 10.777 | 8.781 | 8.456 | 6.097 | 5.373 |
| | BCD(k=2) | 10.797 | **10.775** | 8.786 | 8.454 | 6.097 | 5.372 |
| | OWC-CD | 10.981 | 10.786 | 8.802 | 8.519 | 6.106 | 5.377 |
| | OWC-CD + CD | 10.766 | 10.776 | 8.778 | 8.436 | 6.092 | 5.371 |
| | **OWC-CD + BCD(k=2)** | **10.760** | 10.780 | **8.774** | **8.434** | **6.091** | **5.371** |

Table 3: Table presents the perplexity evaluations for INT2 per-channel quantization of FFN weight with QuIP and FrameQuant, where, GPTQ, CD and BCD are used as subroutines.

| | Method | Gemma-1 7B | Gemma-2 9B | Gemma-2 27B |
|---|---|---|---|---|
| | w16a16 | 10.384 | 10.683 | 8.682 |
| QuIP | GPTQ | 25.941 | 13.695 | 11.909 |
| | CD | 24.025 | 13.172 | **11.479** |
| | BCD(k=2) | **19.167** | **13.144** | 11.547 |
| FrameQuant | GPTQ | 26.262 | 12.941 | 10.958 |
| | CD | 19.726 | 12.748 | 10.785 |
| | BCD(k=2) | **18.242** | **12.674** | **10.608** |

Table 4: Table presents perplexity for GPTQ, CD, BCD for INT3 quantization of FFN layers with AWQ.

| Method | Gemma-1 7B | | Gemma-2 9B | | PaLM2-Otter | |
|---|---|---|---|---|---|---|
| | w3a16 | w3a16g128 | w3a16 | w3a16g128 | w3a16 | w3a16g128 |
| GPTQ + AWQ | 25.695 | 14.539 | 11.331 | 11.158 | 7.381 | 6.519 |
| CD + AWQ | 19.703 | 13.254 | **11.255** | 11.123 | 7.244 | 6.475 |
| **BCD(k=2) + AWQ** | **18.137** | **13.205** | 11.265 | **11.119** | **7.211** | **6.474** |

generation and ranking tasks. For a detailed per-task breakdown of the results please refer to Appendix F.2. These results demonstrate that our models achieve comparable, if not superior, performance to GPTQ across all tasks. Notably, our techniques exhibit significant improvements over GPTQ for smaller models like PaLM2-Gecko and Gemma-1 7B.

**Layer-wise objective.** Finally, in our experiments we also observe that coordinate descent techniques are better at optimizing the layer-wise objective in Equation (1), than GPTQ. For instance, for INT3 per-channel quantization of Gemma-2 9B, the average objective value (relative to all 0's solution) for GPTQ, for the $1^{st}$ feed-forward layer is 0.1449, whereas for CD it is 0.1362, and for BCD(k=2) it is 0.1357 (which translates to $\sim 6\%$ reduction in the objective value).

## 4.2 VERSATILITY OF CDQUANT

A major strength of our algorithm is its versatility. It seamlessly replaces GPTQ in any quantization technique that relies on it. To illustrate this, we focus on the AWQ, QuIP and FrameQuant (the latter two are state-of-the-art PTQ techniques). We demonstrate that our algorithm, when layered on top of the aforementioned algorithms, surpasses the performance of GPTQ layered on top of them. Table 3 presents the results for QuIP, FrameQuant, and Table 4 presents the results for AWQ. It can be seen that both CD and BCD provide $\sim 5\%$ boost in performance for QuIP and FrameQuant. For AWQ, we observe that CD, BCD always outperform GPTQ, for both per-channel and sub-channel quantization (the performance of AWQ on PaLM2-Bison is much worse than not performing AWQ. Hence, in Table 4, we only present results for PaLM2-Otter from the PaLM2 family of models).

Table 5: Runtime (in minutes using 8 H100s) comparison of CDQuant, GPTQ for Gemma-2 27B. Note that FFN1 and FFN1-Gate have the same runtimes, hence we report only FFN1's runtime.

| Config | Method | FFN1 | FFN2 | Attn. |
|--------|--------|------|------|-------|
| | GPTQ | 0.3 | 7.37 | 1.18 |
| w3a16 | CD | 1.23 | 14.1 | 0.51 |
| | BCD(k=2) | 6.03 | 54.94 | 2.11 |
| | GPTQ | 0.3 | 7.35 | 1.18 |
| w4a16 | CD | 1.65 | 17.87 | 0.64 |
| | BCD(k=2) | 14.54 | 133.34 | 4.64 |

Table 6: Table presents the perplexity evaluations for INT3 quantization with BCD wherein the FFN2 may be quantized either with CD or BCD, while rest of the FFN parameters are quantized with BCD.

| Config | Method | Gemma-1 7B | Gemma-2 9B |
|--------|--------|------------|------------|
| w16a16 | | 10.384 | 10.683 |
| w3a16 | CD, FFN2 | 19.709 | 11.288 |
| | BCD(k=2), FFN2 | 18.552 | 11.284 |
| w3a16g128 | CD, FFN2 | 13.512 | 11.179 |
| | BCD(k=2), FFN2 | 13.501 | 11.180 |

## 5 RUNTIME

Table 5 shows the runtime of GPTQ, CD and BCD for quantization of FFN1, FFN2, and attention weights of Gemma-2 27B model using 8 H100s (see Appendix F.4 for 1 GPU runtime). For attention weight quantization, CD is comparable to GPTQ in speed, while BCD is $2\times$ slower. For FFN1 (FFN2) quantization, CD is $5\times$ ($2\times$) slower than GPTQ, whereas BCD is an order of magnitude slower than GPTQ. It can be seen that FFN2 quantization is the most time-consuming step (because the quantization axis is of size $d_{\text{hidden}}$ which is much larger than the quantization axis of other layers). Consequently, CD (BCD) is twice (order of magnitude) as slow as GPTQ for whole-model quantization. We now provide two simple strategies to speed up both CD and BCD.

**Speeding up BCD.** To speed up BCD, we quantize FFN2 with CD, and rest of the layers with BCD (where the quantization axis is of size $d_{\text{model}}$). Results for this setting can be found in Table 6. It can be seen that quantizing FFN2 with CD leads to negligible to no drop in performance especially for the larger, and better Gemma-2 9B. With this setup, BCD's runtime is reduced significantly and is comparable to that of CD.

**Speeding up CD.** We find that CD converges to an optimal solution in much fewer iterations than $d_{\text{in}}$ iterations used in our experiments. Based on this finding, we run CD for lesser number of iterations. As shown in Table 22 in Appendix, even with a drastically reduced iteration count (e.g., $d_{\text{in}}/8$), CD maintains near-optimal performance with negligible quality loss, surpassing the performance of GPTQ. This reduction in iterations allows CD to achieve a runtime faster than/comparable to GPTQ.

## 6 CONCLUSION AND FUTURE WORK

In this work, we developed a coordinate descent framework (CDQuant) for quantization of LLMs. CDQuant is a simple and effective alternative to GPTQ, that consistently outperformed it on PaLM2 models. The simplicity of our algorithm makes it a seamless substitute for GPTQ in various algorithmic contexts where GPTQ currently functions as a sub-routine. Our future work aims to further improve the performance of CDQuant. In particular, we would like to use ideas from CDQuant to improve Quantized Aware Training (QAT). Furthermore, we will focus on developing layer-wise loss functions that are more closely aligned with end-to-end loss, thereby reducing the performance gap between full-precision and quantized models.

Table 7: Table presents *downstream evaluation* (zero-shot) numbers for GPTQ, CD, BCD for INT3, INT4 quantization of FFN weights for the PaLM2 family of models. *Gen*, *Rank* columns correspond to generation and ranking tasks.

| Config | Method | PaLM2-Gecko | | | PaLM2-Otter | | | PaLM2-Bison | | |
|---|---|---|---|---|---|---|---|---|---|---|
| | | Gen. | Rank | Avg. | Gen. | Rank | Avg. | Gen. | Rank | Avg. |
| w16a16 | - | 20.1 | 63.02 | 43.84 | 36.23 | 79.51 | 58.57 | 44.29 | 85.44 | 64.55 |
| w3a16 | OWC | 15.45 | 55.73 | 39.62 | 15.06 | 59.29 | 41.60 | 39.58 | 75.45 | 61.10 |
| | GPTQ | 13.08 | 56.92 | 39.39 | 31.80 | 71.94 | 55.88 | 41.42 | 76.70 | 62.58 |
| | CD | 15.82 | 58.19 | 41.24 | 32.34 | 72.03 | 56.16 | 41.66 | 76.71 | 62.69 |
| | BCD(k=2) | 16.16 | 57.85 | 41.17 | 32.04 | 72.22 | 56.15 | 41.37 | 76.78 | 62.62 |
| w3a16g128 | OWC | 18.54 | 56.95 | 41.59 | 26.62 | 68.09 | 51.50 | 41.54 | 76.51 | 62.52 |
| | GPTQ | 17.96 | 58.30 | 42.16 | 33.43 | 72.98 | 57.16 | 43.47 | 76.65 | 63.38 |
| | CD | 16.81 | 58.10 | 41.58 | 32.73 | 72.62 | 56.66 | 42.95 | 77.03 | 63.4 |
| | BCD(k=2) | 16.33 | 57.81 | 41.22 | 32.13 | 72.67 | 56.46 | 42.67 | 76.85 | 63.18 |
| | OWC-CD | 16.73 | 57.74 | 41.34 | 32.90 | 72.40 | 56.60 | 42.23 | 77.02 | 63.11 |
| | OWC-CD + CD | 16.37 | 58.22 | 41.48 | 33.53 | 72.64 | 56.99 | 43.09 | 76.91 | 63.38 |
| | OWC-CD + BCD(k=2) | 16.47 | 58.45 | 41.66 | 33.71 | 72.73 | 57.12 | 43.07 | 76.70 | 63.25 |
| w4a16 | OWC | 16.07 | 59.42 | 42.08 | 33.76 | 72.22 | 56.83 | 43.59 | 77.13 | 63.71 |
| | GPTQ | 18.77 | 59.5 | 43.21 | 34.03 | 73.53 | 57.73 | 44.16 | 77.54 | 64.19 |
| | CD | 18.20 | 59.95 | 43.25 | 35.01 | 73.42 | 58.06 | 43.97 | 77.55 | 64.12 |
| | BCD(k=2) | 17.89 | 59.64 | 42.94 | 35.10 | 73.31 | 58.02 | 43.95 | 77.77 | 64.24 |
| w4a16g128 | OWC | 19.00 | 60.42 | 43.85 | 34.52 | 72.83 | 57.5 | 44.45 | 77.71 | 64.41 |
| | GPTQ | 19.07 | 60.40 | 43.87 | 35.34 | 73.60 | 58.29 | 44.25 | 77.65 | 64.29 |
| | CD | 20.35 | 60.94 | 44.70 | 36.20 | 73.67 | 58.68 | 44.21 | 77.72 | 64.31 |
| | BCD(k=2) | 20.44 | 60.69 | 44.59 | 36.11 | 73.54 | 58.57 | 44.17 | 77.56 | 64.2 |
| | OWC-CD | 19.80 | 60.39 | 44.15 | 36.33 | 73.62 | 58.71 | 44.75 | 77.70 | 64.52 |
| | OWC-CD + CD | 20.47 | 60.85 | 44.70 | 35.98 | 73.59 | 58.54 | 44.77 | 77.52 | 64.42 |
| | OWC-CD + BCD(k=2) | 20.15 | 60.49 | 44.36 | 36.00 | 73.62 | 58.57 | 44.70 | 77.70 | 64.50 |

Table 8: Table presents *downstream evaluation* (zero-shot) numbers for GPTQ, CD, BCD for INT3, INT4 quantization of FFN weights for both, Gemma-1 and Gemma-2 models. *Gen*, *Rank* columns correspond to generation and ranking tasks.

| Config | Method | Gemma-1 7B | | | Gemma-2 9B | | | Gemma-2 27B | | |
|---|---|---|---|---|---|---|---|---|---|---|
| | | Gen. | Rank | Avg. | Gen. | Rank | Avg. | Gen. | Rank | Avg. |
| w16a16 | | 40.6 | 76.96 | 58.35 | 49.34 | 82.6 | 64.36 | 52.57 | 77.53 | 67.55 |
| w316 | OWC | 0.76 | 43.95 | 26.6 | 45.84 | 73.51 | 62.44 | 47.29 | 75.21 | 64.04 |
| | GPTQ | 19.58 | 58.33 | 40.87 | 45.82 | 73.11 | 62.19 | 49.91 | 75.53 | 65.28 |
| | CD | 27.83 | 69.75 | 50.2 | 46.48 | 73.4 | 62.63 | 50.09 | 75.98 | 65.62 |
| | BCD(k=2) | 28.21 | 70.08 | 50.51 | 46.23 | 73.34 | 62.49 | 50.34 | 76.03 | 65.75 |
| w3a16g128 | OWC | 28.47 | 66.85 | 48.65 | 46.9 | 73.35 | 62.77 | 50.77 | 76.81 | 66.39 |
| | GPTQ | 32.94 | 73.23 | 53.82 | 46.62 | 73.52 | 62.76 | 51.26 | 76.49 | 66.4 |
| | CD | 34.22 | 73.03 | 54.08 | 46.59 | 73.34 | 62.64 | 51.32 | 77.1 | 66.78 |
| | BCD(k=2) | 34.52 | 73.82 | 54.65 | 46.65 | 73.35 | 62.67 | 51.22 | 77.22 | 66.82 |
| | OWC-CD | 32.76 | 72.61 | 53.39 | 46.64 | 73.03 | 62.48 | 50.99 | 76.97 | 66.57 |
| | OWC-CD + CD | 34.3 | 73.8 | 54.57 | 46.7 | 73.46 | 62.75 | 51.55 | 77.07 | 66.86 |
| | OWC-CD + BCD(k=2) | 34.59 | 74.21 | 54.9 | 46.7 | 73.47 | 62.76 | 51.49 | 77.12 | 66.87 |
| w4a16 | OWC | 30.69 | 68.82 | 50.5 | 48.49 | 74.1 | 63.85 | 51.63 | 76.9 | 66.79 |
| | GPTQ | 35.95 | 74.94 | 55.75 | 48.5 | 74.39 | 64.03 | 52.12 | 76.98 | 67.04 |
| | CD | 37.95 | 75.6 | 56.74 | 48.74 | 74.27 | 64.05 | 51.85 | 77.2 | 67.06 |
| | BCD(k=2) | 38.11 | 75.95 | 57 | 48.72 | 74.3 | 64.06 | 51.9 | 77.11 | 67.03 |
| w4a16g128 | OWC | 38.51 | 75.35 | 56.76 | 49.22 | 73.98 | 64.08 | 52.05 | 77.6 | 67.38 |
| | GPTQ | 39.85 | 76.68 | 57.96 | 48.9 | 73.98 | 63.95 | 52.12 | 77.63 | 67.43 |
| | CD | 40.01 | 76.79 | 58.08 | 48.9 | 74.05 | 63.99 | 52.29 | 77.6 | 67.47 |
| | BCD(k=2) | 40.14 | 76.61 | 58.01 | 49.02 | 73.98 | 64 | 52.22 | 77.65 | 67.48 |
| | OWC-CD | 39.44 | 75.99 | 57.43 | 49.26 | 74.21 | 64.23 | 52.15 | 77.75 | 67.51 |
| | OWC-CD + CD | 40.19 | 76.89 | 58.19 | 49.06 | 74.13 | 64.1 | 52.24 | 77.71 | 67.52 |
| | OWC-CD + BCD(k=2) | 39.94 | 76.52 | 57.89 | 49.1 | 74.17 | 64.14 | 52.14 | 77.69 | 67.47 |

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

## A    BROADER IMPACT

We introduce a Coordinate Descent based approach, CDQuant, for compressing Large Language Models. Our method uses only a small amount of data for calibration. We do not foresee any ethical implications arising from the technical aspects of our approach. However, compressing LLMs may give rise to bias effects, a study of which seems essential given the extensive use of LLMs. Our work may be of assistance to such studies. Also, since quantization allows for easier deployment of LLMs, it could have potential societal implications which seem difficult to predict.

## B    LIMITATIONS

- While both CD and BCD outperformed GPTQ in our experiments, BCD achieves slightly better performance than CD. However, BCD is not as fast as CD and can be expensive for large models. In future, we aim to develop techniques to speed up BCD.

- Our algorithms still don't bridge the gap between QAT and PTQ, especially on smaller models. To bridge this gap, we believe one should move away from the $\ell_2^2$ surrogate loss that is being considered by most of the existing work. Instead, we should design surrogate losses that are more closely aligned with end-to-end loss.

## C    RELATED WORK

In this section, we provide a more thorough comparison of our technique with OBQ and GPTQ.

**Detailed comparison with OBQ.**    At a high level, both OBQ, and our technique use greedy strategies to quantize the weights. However, OBQ is extremely slow and doesn't even scale to models with a few billion parameters. The primary reason for this unreasonable slowness is because of a more complicated step involved in OBQ which requires updating all the unquantized coordinates in each step (please refer to equation 2 in Frantar & Alistarh (2022)). Implementing this step in GPUs is extremely slow because of the sheer number of gathers/scatters that need to be performed at each step. In contrast, CDQuant doesn't have this step. At each step of coordinate descent, we only update the single coordinate that leads to the biggest drop in performance. As a result, CDQuant is orders of magnitude faster than OBQ. To quantify this last statement, note that GPTQ paper only ran OBQ for models with a few hundred millions of parameters and showed that OBQ is almost 120x slower than GPTQ (note that this difference will only grow with larger models because OBQ will require even more gathers/scatters). In fact, in the OBQ paper (Frantar & Alistarh, 2022), in Table 6, OBQ takes 65 minutes to quantize a ResNet50 model with 25M parameters. That being said, scaling it to models with billions of parameters seems extremely difficult.

**Detailed comparison with GPTQ.**    GPTQ, as discussed in Section 2, is a heuristic designed to accelerate OBQ by replacing its greedy strategy with a single, predetermined cycle through all coordinates. While this approach improves speed, it often sacrifices accuracy. Although multiple cycles could potentially mitigate this performance drop, our experiments revealed no significant improvement in quality. In contrast, we propose a simple yet effective greedy coordinate descent strategy that achieves superior accuracy while maintaining the computational efficiency of GPTQ.

## D    COMPARISON TO QUANTEASE

In this section, we compare our per-channel quantization results with those of QuantEase, a parallel study to ours. It's important to note that QuantEase does not have a publicly available implementation. So, we implemented the cyclic coordinate descent strategy used by QuantEase, and used the same initialization and regularization strength as our algorithms (although the QuantEase paper doesn't provide these details). We then ran QuantEase for the recommended number of iterations specified in the paper (20 epochs or $T = 20d_{\text{in}}$ iterations). The findings are presented in Table 9. On PaLM2-Gecko, PaLM2-Otter, and PaLM2-Bison, both these approaches have similar performance. These results collectively highlight the potential of coordinate descent algorithms in outperforming GPTQ, especially for low bit quantization.

Table 9: Comparison of QuantEase with CD and BCD for FFN quantization.

| Config | Method | PaLM2-Gecko | PaLM2-Otter | PaLM2-Bison |
|--------|--------|-------------|-------------|-------------|
| | GPTQ | 11.347 | 7.176 | 5.774 |
| w3a16 | QuantEase (epochs=20) | 10.731 | 6.996 | 5.741 |
| | CD | 10.920 | 7.002 | 5.739 |
| | BCD(k=2) | 10.898 | 6.979 | 5.733 |
| | GPTQ | 8.764 | 6.249 | 5.417 |
| w4a16 | QuantEase (epochs=20) | 8.670 | 6.197 | 5.408 |
| | CD | 8.694 | 6.195 | 5.407 |
| | BCD(k=2) | 8.691 | 6.192 | 5.405 |

## E  CDQUANT FOR SUB-CHANNEL QUANTIZATION

To simplify the explanation in this section, we introduce a slightly modified notation. Given a weight vector $\mathbf{w}$, we represent it's sub-channel quantization as $\mathbf{a} \odot \mathbf{q} + \mathbf{b}$, where $\odot$ is the elementwise multiplication, and $\mathbf{a}, \mathbf{b} \in \mathbb{R}^{d_{\text{in}}}$ are the scale, and bias parameters that satisfy the following constraints: $\mathbf{a}_k = \mathbf{a}_l, \mathbf{b}_k = \mathbf{b}_l$, for any two indices $k, l$ that fall in the same group. With this notation, for any given $\mathbf{a}, \mathbf{b}$, the optimization problem in Equation (3) can be rewritten as

$$\min_{\mathbf{q}} \|X(\mathbf{w} - \mathbf{a} \odot \mathbf{q} - \mathbf{b})\|_2^2. \tag{5}$$

Letting $D_{\mathbf{a}} = \text{diag}(\mathbf{a}), \tilde{X} = X D_{\mathbf{a}}, \tilde{\mathbf{w}} = D_{\mathbf{a}}^{-1}\mathbf{w}, \tilde{\mathbf{b}} = D_{\mathbf{a}}^{-1}\mathbf{b}$, the above problem can be further rewritten as

$$\min_{\mathbf{q}} \|\tilde{X}(\tilde{\mathbf{w}} - \mathbf{q} - \tilde{\mathbf{b}})\|_2^2. \tag{6}$$

Observe that this problem is the same as the per-channel quantization problem described in Equation (1), but with modified parameters $\tilde{X}, \tilde{\mathbf{w}}, \tilde{\mathbf{b}}$. So extending CDQuant to sub-channel quantization simply involves running Algorithms 1, 2 with these modified parameters. Algorithms 4, 5 present the resulting algorithms.

---

**Algorithm 4** Greedy Coordinate Descent (CD)

---

1: **Input:** $T$ - coordinate descent steps, $X$ - input data matrix, $\mathbf{w}$ - vector to be quantized, $a$ - scale, $b$ - bias, $\mathbf{q}_0$ - initial estimate
2: $\tilde{X} \leftarrow X\text{diag}(\mathbf{a}), \tilde{\mathbf{w}} = \text{diag}(\mathbf{a})^{-1}\mathbf{w}, \tilde{\mathbf{b}} = \text{diag}(\mathbf{a})\mathbf{b}$
3: Compute Hessian $H$ as: $H \leftarrow \tilde{X}^T\tilde{X}$
4: Compute gradient $\mathbf{g}$ as: $\mathbf{g} \leftarrow 2H(\mathbf{q}_0 - (\tilde{\mathbf{w}} - \tilde{\mathbf{b}}))$
5: **for** $t \in [1 : T]$ **do**
6:     Find the coordinate that leads to the largest reduction in loss

$$i^*, r^* = \underset{i \in \{0, 1, \ldots d_{\text{in}}-1\}, r \in \{0, 1, \ldots 2^c-1\}}{\arg\min} (r - \mathbf{q}_{t-1,i})^2 H_{i,i} + (r - \mathbf{q}_{t-1,i})\mathbf{g}_i$$

7:     Update gradient $\mathbf{g}$ as
$$\mathbf{g} \leftarrow \mathbf{g} + 2(r^* - \mathbf{q}_{t-1,i^*})H_{i^*,.},$$
    where $H_{i^*,.}$ is the $i^*$ column of $H$
8:     Update $\mathbf{q}_{t-1}$ as
$$\mathbf{q}_t \leftarrow \mathbf{q}_{t-1} + (r^* - \mathbf{q}_{t-1,i^*})\mathbf{e}_{i^*},$$
    where $\mathbf{e}_{i^*}$ is the standard basis vector with 1 in $i^*$ position and 0 everywhere else
9: **end for**

---

## F  ADDITIONAL EXPERIMENTAL RESULTS

This section presents additional experimental results:

---

**Algorithm 5** Block Coordinate Descent with Random Blocks (BCD)

---

1: **Input:** $T$ - coordinate descent steps, $k$ - block size, $X$ - input data matrix, $\mathbf{w}$ - vector to be quantized, $a$ - scale, $b$ - bias, $\mathbf{q}_0$ - initial estimate
2: $\tilde{X} \leftarrow X\text{diag}(\mathbf{a}), \tilde{\mathbf{w}} = \text{diag}(\mathbf{a})^{-1}\mathbf{w}, \tilde{\mathbf{b}} = \text{diag}(\mathbf{a})\mathbf{b}$
3: Compute Hessian $H$ as: $H \leftarrow \tilde{X}^T\tilde{X}$
4: Compute gradient $\mathbf{g}$ as: $\mathbf{g} \leftarrow 2H(\mathbf{q}_0 - (\tilde{\mathbf{w}} - \tilde{\mathbf{b}}))$
5: **for** $t \in [1 : T]$ **do**
6:     Randomly partition the set $\{0, 1, \ldots d_{\text{in}} - 1\}$ into $d_{\text{in}}/k$ blocks, each of size $k$
7:     Find the block that leads to the largest reduction in loss

$$i^*, r^* = \underset{\substack{i\in\{0,1,\ldots d_{\text{in}}/k-1\}, \\ r\in\{0,1,\ldots 2^c-1\}^k}}{\arg\min} (r - \mathbf{q}_{t-1,i})^T H_{i,i}(r - \mathbf{q}_{t-1,i}) + (r - \mathbf{q}_{t-1,i})^T \mathbf{g}_i,$$

    where $\mathbf{q}_{t-1,i}, H_{i,i}$ are the sub-vector, sub-matrix of $\mathbf{q}_{t-1}, H$ corresponding to block $i$.
8:     Update gradient $\mathbf{g}$ as
$$\mathbf{g} \leftarrow \mathbf{g} + 2H_{i^*,\cdot}(r^* - \mathbf{q}_{t-1,i^*}),$$
9:     Update $\mathbf{q}_{t-1}$ as
$$\mathbf{q}_{t-1,i^*} \leftarrow r^*, \quad \mathbf{q}_t \leftarrow \mathbf{q}_{t-1},$$
10: **end for**

---

1. In Appendix F.1, we present the effects of a larger block size and multiple epochs on BCD.

2. In Appendix F.2, we present a detailed breakdown of downstream evaluation results on various datasets.

3. In Appendix F.3, we present results for the setting where we quantize both attention and FFN layers.

4. In Appendix F.4, we present runtime numbers for CD and BCD algorithms

## F.1 ADDITIONAL ABLATIONS FOR BCD

We investigate the effects of a large block size and multiple epochs on BCD, where each epoch corresponds to $d_{\text{in}}$ iterations. Results for this experiment are presented in Table 10. For BCD with $k = 2$, additional epochs enhance performance. However, for BCD with $k = 4$, multiple epochs appear to lead to overfitting, indicating a need for stronger regularization.

Table 10: Table presents the perplexity evaluations BCD for INT2 quantization of FFN weights when run with a larger block size and multiple epochs.

| Method | Epochs | Gemma-1 7B | Gemma-2 9B | Gemma-2 27B | PaLM2-Otter | PaLM2-Bison |
|---|---|---|---|---|---|---|
| w16a16 | | 10.384 | 10.683 | 8.682 | 5.984 | 5.298 |
| w2a16g128 | | | | | | |
| GPTQ | - | 375.153 | 13.785 | 12.181 | 10.816 | 7.230 |
| CD | 1 | 75.55 | 13.709 | 11.966 | 9.917 | 7.123 |
| BCD(k=2) | 1 | **68.732** | **13.662** | **11.873** | **9.822** | **7.094** |
| | 2 | 67.766 | 13.665 | 11.888 | 9.760 | **7.088** |
| | 3 | 72.330 | 13.674 | 11.878 | 9.719 | 7.094 |
| BCD(k=4) | 1 | 68.785 | **13.645** | **11.857** | **9.708** | 7.099 |
| | 2 | 67.390 | 13.640 | 11.867 | 9.752 | 7.099 |
| | 3 | 67.590 | 13.664 | 11.869 | 9.749 | 7.099 |

## F.2 DETAILED DOWNSTREAM EVALUATION RESULTS

Table 11: Table presents *downstream evaluation* (zero-shot) numbers of PaLM2-Gecko - for GPTQ, CD, BCD for INT3, INT4 quantization of FFN weights.

| Config | Method | PaLM2-Gecko | | | | | | | | | |
| --- | --- | --- | --- | --- | --- | --- | --- | --- | --- | --- | --- |
| | | NatualQ. | SQuAD | TriviaQA | WebQ | ARC-c | ARC-e | BoolQ | HellaSwag | PIQA | WinoGrande |
| w16a16 | - | 4.68 | 43.17 | 27.46 | 5.07 | 35.24 | 65.66 | 60.24 | 60.69 | 74.7 | 61.48 |
| w3a16 | OWC | 2.55 | 47.57 | 9.81 | 1.87 | 29.69 | 56.73 | 64.8 | 53.28 | 69.59 | 60.3 |
| | GPTQ | 3.35 | 44.13 | 3.97 | 0.89 | 30.2 | 62.12 | 63.79 | 56.33 | 71.22 | 57.85 |
| | CD | 3.55 | 45.71 | 11.32 | 2.71 | 30.63 | 60.56 | 66.45 | 57.2 | 71.87 | 62.43 |
| | **BCD(k=2)** | 3.63 | 47.04 | 11.22 | 2.76 | 30.29 | 60.52 | 65.32 | 57.33 | 71.82 | 61.8 |
| w3a16g128 | OWC | 3.68 | 55.38 | 12.5 | 2.61 | 31.66 | 60.14 | 66.12 | 53.19 | 71.71 | 58.88 |
| | GPTQ | 4.02 | 50.04 | 14.64 | 3.15 | 32.42 | 61.91 | 66.64 | 56.05 | 73.01 | 59.75 |
| | CD | 3.38 | 52.01 | 9.62 | 2.21 | 31.23 | 63.13 | 65.9 | 55.98 | 72.47 | 59.91 |
| | BCD(k=2) | 3.1 | 50.95 | 9.36 | 1.92 | 31.14 | 62.25 | 65.41 | 56.16 | 72.91 | 58.96 |
| | OWC-CD | 3.6 | 51.69 | 9.33 | 2.31 | 31.83 | 61.74 | 65.02 | 56.23 | 72.2 | 59.43 |
| | OWC-CD + CD | 3.82 | 50.39 | 9.5 | 1.77 | 32.51 | 62.25 | 66.33 | 56.91 | 71.27 | 60.06 |
| | **OWC-CD + BCD(k=2)** | 3.63 | 49.87 | 10.26 | 2.12 | 32.76 | 62.88 | 66.24 | 56.79 | 71.82 | 60.22 |
| w4a16 | OWC | 4.02 | 46.96 | 10.63 | 2.66 | 33.19 | 61.99 | 66.7 | 59.26 | 73.67 | 61.72 |
| | GPTQ | 4.46 | 48.69 | 18.84 | 3.1 | 34.47 | 62.96 | 64.77 | 60.09 | 73.61 | 61.09 |
| | CD | 4.13 | 48.88 | 16.64 | 3.15 | 33.53 | 63.89 | 66.79 | 60.31 | 73.78 | 61.4 |
| | **BCD(k=2)** | 4.24 | 48.31 | 15.73 | 3.3 | 33.28 | 63.34 | 66.39 | 60.26 | 73.56 | 61.01 |
| w4a16g128 | OWC | 4.99 | 46.1 | 20.62 | 4.28 | 34.73 | 64.48 | 67.83 | 59.91 | 74.32 | 61.25 |
| | GPTQ | 5.18 | 49.79 | 18.17 | 3.15 | 35.15 | 64.6 | 66.24 | 60.21 | 74.32 | 61.88 |
| | CD | 5.73 | 46.96 | 24.17 | 4.53 | 36.18 | 66.04 | 67.8 | 60.7 | 73.67 | 61.25 |
| | BCD(k=2) | 5.79 | 47.28 | 24.11 | 4.58 | 36.01 | 65.7 | 67.06 | 60.73 | 73.61 | 61.01 |
| | OWC-CD | 5.54 | 47.24 | 21.71 | 4.72 | 34.3 | 64.73 | 68.01 | 60.72 | 73.56 | 61.01 |
| | OWC-CD + CD | 5.62 | 47.6 | 24 | 4.68 | 35.49 | 66.16 | 67.06 | 60.78 | 74.1 | 61.48 |
| | **OWC-CD + BCD(k=2)** | 5.57 | 47.17 | 23.05 | 4.82 | 35.24 | 65.82 | 66.02 | 60.9 | 73.23 | 61.72 |

Table 12: Table presents *downstream evaluation* (zero-shot) numbers of PaLM2-Otter - for GPTQ, CD, BCD for INT3, INT4 quantization of FFN weights.

| Config | Method | PaLM2-Otter | | | | | | | | | |
| --- | --- | --- | --- | --- | --- | --- | --- | --- | --- | --- | --- |
| | | NatualQ. | SQuAD | TriviaQA | WebQ | ARC-c | ARC-e | BoolQ | HellaSwag | PIQA | WinoGrande |
| w16a16 | - | 12.85 | 65.27 | 58.34 | 8.46 | 51.79 | 76.09 | 81.8 | 79.3 | 79.6 | 72.22 |
| w3a16 | OWC | 2.27 | 49.45 | 8.02 | 0.49 | 41.98 | 60.14 | 48.78 | 66.91 | 71.87 | 66.06 |
| | GPTQ | 9.09 | 64.53 | 46.38 | 7.19 | 49.83 | 73.99 | 81.96 | 76.87 | 78.89 | 70.09 |
| | CD | 10.72 | 64.01 | 48.03 | 6.59 | 50.77 | 76.6 | 81.01 | 75.77 | 77.97 | 70.09 |
| | **BCD(k=2)** | 10.55 | 64.31 | 47.25 | 6.05 | 50.85 | 76.43 | 81.41 | 75.91 | 78.45 | 70.24 |
| w3a16g128 | OWC | 6.18 | 66.37 | 30.48 | 3.44 | 47.44 | 70.16 | 69.63 | 74.61 | 76.88 | 69.85 |
| | GPTQ | 10.17 | 66.53 | 49.29 | 7.73 | 51.37 | 75.93 | 82.35 | 77 | 79.11 | 72.14 |
| | CD | 7.17 | 68.73 | 49.42 | 5.61 | 50.26 | 76.18 | 82.39 | 76.79 | 78.13 | 71.98 |
| | BCD(k=2) | 6.68 | 68.18 | 48.39 | 5.27 | 50.26 | 75.93 | 81.41 | 76.84 | 79 | 72.61 |
| | OWC-CD | 8.98 | 68.37 | 48.33 | 5.91 | 50.51 | 75.46 | 81.35 | 76.83 | 78.18 | 72.06 |
| | OWC-CD + CD | 9.72 | 67.87 | 49.87 | 6.64 | 51.37 | 76.68 | 81.59 | 76.78 | 77.91 | 71.51 |
| | **OWC-CD + BCD(k=2)** | 9.94 | 68.26 | 49.91 | 6.74 | 50.85 | 76.22 | 81.68 | 76.99 | 78.89 | 71.74 |
| w4a16 | OWC | 11.19 | 66.67 | 49.56 | 7.63 | 50 | 73.91 | 80 | 77.36 | 79.11 | 72.93 |
| | GPTQ | 12.27 | 66.65 | 48.72 | 8.46 | 51.71 | 75.84 | 82.02 | 78.2 | 80.09 | 73.32 |
| | CD | 12.52 | 66.85 | 53.16 | 7.53 | 52.3 | 76.43 | 81.28 | 78.3 | 79.6 | 72.61 |
| | **BCD(k=2)** | 12.33 | 67 | 53.45 | 7.63 | 51.96 | 75.97 | 80.92 | 78.47 | 80.47 | 72.06 |
| w4a16g128 | OWC | 11.33 | 65.98 | 53.57 | 7.19 | 49.66 | 74.92 | 82.17 | 78.22 | 79.54 | 72.45 |
| | GPTQ | 11.5 | 65.81 | 56.36 | 7.68 | 51.71 | 76.52 | 82.08 | 78.86 | 79.27 | 73.16 |
| | CD | 11.72 | 66.85 | 57.63 | 8.61 | 51.19 | 76.6 | 82.51 | 78.89 | 79.49 | 73.32 |
| | BCD(k=2) | 11.66 | 66.73 | 57.5 | 8.56 | 51.19 | 76.3 | 82.29 | 78.88 | 79.27 | 73.32 |
| | OWC-CD | 12.27 | 66.28 | 58.21 | 8.56 | 51.62 | 76.85 | 82.32 | 78.85 | 79.49 | 72.61 |
| | OWC-CD + CD | 11.63 | 66.28 | 58.06 | 7.92 | 51.62 | 76.94 | 82.45 | 78.72 | 79.11 | 72.69 |
| | **OWC-CD + BCD(k=2)** | 11.72 | 65.96 | 58.29 | 8.02 | 51.79 | 76.77 | 82.17 | 78.8 | 79.49 | 72.69 |

Table 13: Table presents *downstream evaluation* (zero-shot) numbers of PaLM2-Bison - for GPTQ, CD, BCD for INT3, INT4 quantization of FFN weights.

| Config | Method | PaLM2-Bison | | | | | | | | | |
|---|---|---|---|---|---|---|---|---|---|---|---|
| | | NatualQ. | SQuAD | TriviaQA | WebQ | ARC-c | ARC-e | BoolQ | HellaSwag | PIQA | WinoGrande |
| w16a16 | - | 21.36 | 72.92 | 70 | 12.89 | 55.38 | 81.27 | 87.86 | 82.72 | 82.97 | 78.14 |
| w3a16 | OWC | 15.87 | 73.01 | 58.86 | 10.58 | 52.47 | 76.43 | 86.91 | 79.76 | 81.01 | 76.09 |
| | GPTQ | 17.31 | 71.25 | 65.53 | 11.56 | 54.44 | 79.25 | 87.22 | 80.78 | 81.61 | 76.87 |
| | CD | 17.84 | 71.24 | 64.76 | 12.8 | 54.44 | 79.17 | 87.58 | 80.79 | 82.1 | 76.16 |
| | **BCD(k=2)** | 17.81 | 71.3 | 64.29 | 12.06 | 53.5 | 79.08 | 87.65 | 80.91 | 82.21 | 77.35 |
| w3a16g128 | OWC | 15.98 | 71.28 | 68.02 | 10.88 | 53.58 | 79.55 | 86.73 | 80.9 | 81.61 | 76.72 |
| | GPTQ | 18.95 | 71.49 | 69.67 | 13.78 | 53.24 | 79.84 | 86.79 | 81.05 | 82.05 | 76.95 |
| | CD | 18.31 | 71.81 | 68.56 | 13.14 | 53.41 | 79.8 | 87.55 | 81.31 | 82.43 | 77.66 |
| | BCD(k=2) | 18.53 | 71.55 | 67.62 | 12.99 | 53.58 | 79.59 | 87.49 | 81.14 | 82.81 | 76.48 |
| | OWC-CD | 17.26 | 71.79 | 67.53 | 12.35 | 54.69 | 80.85 | 85.72 | 81.02 | 81.88 | 77.98 |
| | OWC-CD + CD | 18.75 | 72.05 | 68.35 | 13.19 | 54.1 | 80.18 | 86.36 | 80.86 | 82.48 | 77.51 |
| | **OWC-CD + BCD(k=2)** | 19.36 | 71.47 | 68.45 | 12.99 | 53.58 | 80.47 | 86.02 | 81.15 | 82.26 | 76.72 |
| w4a16 | OWC | 20 | 72.51 | 69.34 | 12.5 | 54.61 | 80.39 | 87.06 | 81.5 | 82.05 | 77.19 |
| | GPTQ | 21.25 | 72.39 | 70.6 | 12.4 | 55.2 | 80.64 | 87.58 | 81.98 | 82.26 | 77.58 |
| | CD | 21.5 | 71.57 | 70.04 | 12.8 | 55.29 | 80.56 | 88.07 | 81.78 | 82.59 | 77.03 |
| | **BCD(k=2)** | 21.16 | 71.92 | 70.01 | 12.7 | 55.63 | 81.19 | 87.77 | 81.82 | 82.81 | 77.43 |
| w4a16g128 | OWC | 21.22 | 73.06 | 70.07 | 13.44 | 55.8 | 80.47 | 87.77 | 82.08 | 82.64 | 77.51 |
| | GPTQ | 20.72 | 73.17 | 70.12 | 12.99 | 55.38 | 80.89 | 87.55 | 82.14 | 82.59 | 77.35 |
| | CD | 21.8 | 73.37 | 69.06 | 12.6 | 54.78 | 80.72 | 87.25 | 82.16 | 83.19 | 78.22 |
| | BCD(k=2) | 21.5 | 73.06 | 69.34 | 12.8 | 55.29 | 80.98 | 87.31 | 82.14 | 82.54 | 77.11 |
| | OWC-CD | 21.36 | 73.85 | 70.07 | 13.73 | 55.12 | 81.23 | 87.71 | 82.27 | 82.59 | 77.27 |
| | OWC-CD + CD | 21.41 | 73.32 | 71.46 | 12.89 | 54.18 | 81.4 | 87.28 | 82.17 | 82.37 | 77.74 |
| | **OWC-CD + BCD(k=2)** | 21.19 | 73.24 | 71.64 | 12.75 | 55.03 | 81.52 | 87.28 | 82.06 | 82.64 | 77.66 |

Table 14: Table presents *downstream evaluation* (zero-shot) numbers of Gemma-1 7B - for GPTQ, CD, BCD for INT3, INT4 quantization of FFN weights.

| Config | Method | Gemma-1 7B | | | | | | | | | |
|---|---|---|---|---|---|---|---|---|---|---|---|
| | | NaturalQ | SQuAD | TriviaQA | WebQ | ARC-c | ARC-e | BoolQ | HellaSwag | PIQA | WinoGrande |
| w16a16 | | 15.04 | 71.94 | 62.38 | 13.04 | 43.69 | 65.53 | 80.43 | 78.55 | 80.79 | 72.14 |
| w3l6 | OWC | 0.03 | 2.87 | 0.15 | 0 | 30.55 | 38.8 | 63.09 | 29 | 52.61 | 48.86 |
| | GPTQ | 2.24 | 58.68 | 14.35 | 3.05 | 35.15 | 53.16 | 63.76 | 53.84 | 64.58 | 59.91 |
| | CD | 6.04 | 68.77 | 32.22 | 4.28 | 41.04 | 62.37 | 76.51 | 68.95 | 74.76 | 67.01 |
| | BCD(k=2) | 6.76 | 69.03 | 33.32 | 3.74 | 40.7 | 62.21 | 76.91 | 69.1 | 75.24 | 68.11 |
| w3a16g128 | OWC | 6.51 | 71.04 | 30.86 | 5.46 | 38.14 | 60.1 | 70.7 | 64.51 | 75.3 | 63.85 |
| | GPTQ | 10.03 | 71.66 | 42.25 | 7.82 | 41.64 | 64.06 | 80.83 | 72.37 | 78.18 | 69.38 |
| | CD | 10.08 | 71.94 | 46.39 | 8.46 | 40.7 | 62.75 | 80.43 | 73.88 | 78.56 | 67.64 |
| | BCD(k=2) | 10.78 | 71.99 | 46.65 | 8.66 | 42.32 | 63.01 | 81.25 | 74.17 | 78.18 | 69.46 |
| | OWC-CD | 9.7 | 71.62 | 41.44 | 8.27 | 41.47 | 64.44 | 79.94 | 72.19 | 77.53 | 67.32 |
| | OWC-CD + CD | 10.66 | 71.78 | 46.73 | 8.02 | 42.41 | 64.73 | 80.46 | 74.29 | 78.94 | 67.64 |
| | OWC-CD + BCD(k=2) | 10.69 | 71.84 | 47.05 | 8.76 | 42.41 | 65.36 | 80.03 | 74.47 | 79.49 | 68.9 |
| w4a16 | OWC | 7.09 | 63.51 | 44.98 | 7.19 | 41.38 | 59.34 | 77.06 | 67.29 | 73.29 | 63.85 |
| | GPTQ | 10.66 | 70.95 | 52.63 | 9.55 | 44.11 | 64.56 | 79.39 | 75.9 | 79.16 | 70.56 |
| | CD | 12.71 | 70.41 | 57.39 | 11.27 | 43.69 | 65.11 | 79.66 | 75.98 | 79.98 | 70.17 |
| | BCD(k=2) | 12.44 | 70.5 | 57.55 | 11.96 | 45.14 | 65.4 | 79.79 | 76.66 | 79.54 | 71.03 |
| w4a16g128 | OWC | 13.55 | 71.61 | 56.09 | 12.8 | 42.49 | 63.85 | 80.92 | 76.77 | 79.22 | 70.32 |
| | GPTQ | 13.88 | 71.94 | 59.45 | 14.12 | 43.52 | 66.12 | 81.96 | 77.54 | 79.98 | 71.11 |
| | CD | 15.26 | 71.73 | 59.81 | 13.24 | 45.05 | 65.45 | 81.25 | 77.52 | 80.58 | 70.88 |
| | BCD(k=2) | 15.54 | 71.94 | 59.83 | 13.24 | 43.69 | 64.6 | 80.89 | 77.68 | 81.23 | 71.43 |
| | OWC-CD | 14.43 | 72.08 | 58.69 | 12.55 | 43.52 | 64.94 | 80.46 | 77.46 | 79.49 | 70.64 |
| | OWC-CD + CD | 14.82 | 72.32 | 59.98 | 13.63 | 45.05 | 65.82 | 80.43 | 77.46 | 80.96 | 71.43 |
| | OWC-CD + BCD(k=2) | 14.93 | 72.39 | 59.49 | 12.94 | 44.45 | 66.08 | 80.21 | 77.6 | 80.03 | 70.8 |

Table 15: Table presents *downstream evaluation* (zero-shot) numbers of Gemma-2 9B - for GPTQ, CD, BCD for INT3, INT4 quantization of FFN weights.

| Config | Method | Gemma-2 9B | | | | | | | | | |
|---|---|---|---|---|---|---|---|---|---|---|---|
| | | NaturalQ | SQuAD | TriviaQA | WebQ | ARC-c | ARC-e | BoolQ | HellaSwag | PIQA | WinoGrande |
| w16a16 | | 26.76 | 72.06 | 76.83 | 21.7 | 58.96 | 77.57 | 83.33 | 77.31 | 81.12 | 67.96 |
| w316 | OWC | 21.02 | 74.18 | 69.76 | 18.41 | 56.23 | 76.52 | 84.37 | 75.24 | 80.79 | 67.88 |
| | GPTQ | 21.22 | 74 | 69.99 | 18.06 | 57 | 77.19 | 82.35 | 74.96 | 80.69 | 66.46 |
| | CD | 22.24 | 73.41 | 70.67 | 19.59 | 56.91 | 77.74 | 84.13 | 75.27 | 79.82 | 66.61 |
| | BCD(k=2) | 21.77 | 73.33 | 71 | 18.8 | 57.08 | 77.69 | 83.18 | 75.04 | 80.03 | 67.01 |
| w3a16g128 | OWC | 23.1 | 73.42 | 71.79 | 19.29 | 56.31 | 76.6 | 83.94 | 75.28 | 80.74 | 67.25 |
| | GPTQ | 22.33 | 72.77 | 71.68 | 19.69 | 56.66 | 77.44 | 83.46 | 75.19 | 80.74 | 67.64 |
| | CD | 22.8 | 71.94 | 72.24 | 19.39 | 56.74 | 77.36 | 84.13 | 75.27 | 80.14 | 66.38 |
| | BCD(k=2) | 23.02 | 72.3 | 71.99 | 19.29 | 56.57 | 77.06 | 84.37 | 75.36 | 80.3 | 66.46 |
| | OWC-CD | 22.55 | 72.69 | 72.13 | 19.19 | 55.46 | 76.85 | 82.81 | 75.36 | 80.79 | 66.93 |
| | OWC-CD + CD | 22.99 | 72.42 | 72.28 | 19.09 | 56.91 | 77.36 | 82.75 | 75.48 | 80.52 | 67.72 |
| | OWC-CD + BCD(k=2) | 23.27 | 72.39 | 72.23 | 18.9 | 57 | 77.31 | 83.27 | 75.49 | 80.41 | 67.32 |
| w4a16 | OWC | 25.48 | 73.21 | 74.93 | 20.32 | 58.45 | 76.81 | 83.94 | 76.58 | 80.69 | 68.11 |
| | GPTQ | 25.29 | 72.9 | 74.65 | 21.16 | 58.7 | 77.27 | 83.55 | 76.6 | 81.23 | 68.98 |
| | CD | 25.57 | 73.03 | 75.42 | 20.92 | 58.36 | 77.61 | 83.79 | 76.75 | 80.41 | 68.67 |
| | BCD(k=2) | 25.6 | 72.86 | 75.35 | 21.06 | 58.62 | 77.4 | 83.73 | 76.74 | 80.69 | 68.59 |
| w4a16g128 | OWC | 26.26 | 73.33 | 75.73 | 21.56 | 58.7 | 77.4 | 83.33 | 76.76 | 80.85 | 66.85 |
| | GPTQ | 26.12 | 72.53 | 75.75 | 21.21 | 58.53 | 77.65 | 83.06 | 76.72 | 81.28 | 66.61 |
| | CD | 26.23 | 72.57 | 75.83 | 20.96 | 58.28 | 77.53 | 83.15 | 76.83 | 81.12 | 67.4 |
| | BCD(k=2) | 26.4 | 72.62 | 75.94 | 21.11 | 58.36 | 77.4 | 83.09 | 76.72 | 80.85 | 67.48 |
| | OWC-CD | 26.15 | 73.46 | 75.88 | 21.56 | 59.3 | 77.74 | 82.81 | 76.72 | 80.96 | 67.72 |
| | OWC-CD + CD | 26.15 | 72.7 | 75.98 | 21.41 | 58.62 | 77.78 | 82.94 | 76.86 | 81.07 | 67.48 |
| | OWC-CD + BCD(k=2) | 26.32 | 72.75 | 76.13 | 21.21 | 59.13 | 77.48 | 83 | 76.92 | 81.01 | 67.48 |

Table 16: Table presents *downstream evaluation* (zero-shot) numbers of Gemma-2 27B - for GPTQ, CD, BCD for INT3, INT4 quantization of FFN weights.

| Config | Method | Gemma-2 27B | | | | | | | | | |
|---|---|---|---|---|---|---|---|---|---|---|---|
| | | NaturalQ | SQuAD | TriviaQA | WebQ | ARC-c | ARC-e | BoolQ | HellaSwag | PIQA | WinoGrande |
| w16a16 | | 30.42 | 75.04 | 81.5 | 23.33 | 60.49 | 78.96 | 83.18 | 82.24 | 84.6 | 75.69 |
| w316 | OWC | 23.3 | 73.28 | 73.2 | 19.39 | 56.66 | 77.53 | 81.01 | 79.82 | 83.24 | 73.01 |
| | GPTQ | 26.09 | 75.08 | 76.62 | 21.85 | 57.51 | 78.96 | 77.4 | 80.57 | 83.73 | 74.98 |
| | CD | 26.51 | 75.57 | 77.21 | 21.06 | 59.13 | 79.76 | 78.29 | 80.1 | 83.95 | 74.66 |
| | BCD(k=2) | 26.7 | 75.67 | 77.34 | 21.65 | 58.45 | 79.34 | 78.47 | 79.93 | 83.41 | 76.56 |
| w3a16g128 | OWC | 26.7 | 75.95 | 78.61 | 21.8 | 58.45 | 79.63 | 83.55 | 80.49 | 82.75 | 76.01 |
| | GPTQ | 27.48 | 75.97 | 78.72 | 22.88 | 59.56 | 80.05 | 80.28 | 81.03 | 83.68 | 74.35 |
| | CD | 27.98 | 76.15 | 78.59 | 22.54 | 59.73 | 79.88 | 81.77 | 80.99 | 84.11 | 76.09 |
| | BCD(k=2) | 27.78 | 75.99 | 78.62 | 22.49 | 59.56 | 80.01 | 81.99 | 81.08 | 84.11 | 76.56 |
| | OWC-CD | 26.87 | 76.03 | 79 | 22.05 | 58.96 | 80.18 | 81.28 | 80.94 | 83.79 | 76.64 |
| | OWC-CD + CD | 27.87 | 76.36 | 79.62 | 22.34 | 59.3 | 80.05 | 81.59 | 80.97 | 83.79 | 76.72 |
| | OWC-CD + BCD(k=2) | 27.7 | 76.45 | 79.38 | 22.44 | 59.73 | 80.05 | 81.83 | 80.98 | 83.57 | 76.56 |
| w4a16 | OWC | 29 | 75.51 | 80.05 | 21.95 | 59.39 | 79.04 | 81.13 | 81.93 | 84.39 | 75.53 |
| | GPTQ | 29.5 | 75.7 | 80.41 | 22.88 | 59.47 | 78.7 | 81.62 | 81.91 | 84.33 | 75.85 |
| | CD | 29.36 | 75.23 | 80.23 | 22.59 | 59.9 | 79.67 | 81.53 | 81.84 | 84.49 | 75.77 |
| | BCD(k=2) | 29.42 | 75.34 | 80.31 | 22.54 | 59.3 | 79.46 | 81.83 | 81.72 | 84.44 | 75.93 |
| w4a16g128 | OWC | 29.25 | 75.41 | 80.91 | 22.64 | 59.98 | 79.46 | 83.76 | 82.14 | 84.22 | 76.01 |
| | GPTQ | 29.53 | 75.61 | 80.85 | 22.49 | 60.41 | 79.59 | 83.61 | 82.14 | 84.66 | 75.37 |
| | CD | 29.58 | 75.38 | 81.05 | 23.13 | 59.98 | 79.42 | 83.73 | 82.08 | 84.44 | 75.93 |
| | BCD(k=2) | 29.61 | 75.44 | 80.91 | 22.93 | 60.32 | 79.71 | 83.85 | 82.06 | 84.44 | 75.53 |
| | OWC-CD | 29.11 | 75.93 | 80.85 | 22.69 | 60.75 | 79.67 | 83.33 | 82.2 | 84.87 | 75.69 |
| | OWC-CD + CD | 29.36 | 75.66 | 80.96 | 22.98 | 59.9 | 79.59 | 83.82 | 82.05 | 85.15 | 75.77 |
| | OWC-CD + BCD(k=2) | 29.34 | 75.41 | 80.86 | 22.93 | 59.9 | 79.84 | 83.94 | 82.14 | 84.93 | 75.37 |

## F.3 ATTENTION + FFN QUANTIZATION

Before presenting the results for quantization of both attention and FFN layers, we first explain why quantizing the attention weights does not yield inference latency gains.

### F.3.1 ATTENTION QUANTIZATION DOESN'T YIELD LATENCY GAINS

In Table 17, we present the end-to-end prefix and decode latency for Gemma-2 27B with three different prefix lengths. The latencies are obtained by running Gemma-2 27B on 4 TPUv5 chips with a per TPU batch size of 4 and no model parallelism.

**Prefix phase.** During the prefix phase, neither FFN nor attention quantization yielded significant latency improvements, likely due to the compute-bound nature of the prefix.

**Decode phase.** In the decode phase, quantizing attention weights from 16 bits to 8 bits provided marginal gains, as attention compute is primarily memory-bound by the KV cache size, which typically exceeds attention weight size for moderate-to-large context lengths. Further reduction to 4-bit quantization offered negligible latency improvement while incurring a non-trivial drop in quality (Table 18 and Table 2), thus questioning the utility of attention weight quantization. Conversely, FFN layers, processing only a single token activation during decoding, are memory-bound by the significantly larger FFN weights (9x the size of attention weights in Gemma-2 27B). Consequently, FFN weight quantization effectively reduces memory transfers and improves latency. We observed a $32\% - 35\%$ latency reduction when quantizing from 16 bits to 8 bits, with an additional $10\%$ reduction when using 4-bit quantization.

These results clearly highlight that quantization of attention weights doesn't yield any inference latency gains.

Table 17: Table presents the prefix and decode time inference latency of Gemma-2 27B with three different prefix lengths. In the Attention-only setting, only the attention parameters are quantized and rest of the model is in 16 bits. Similarly, in the FFN-only setting, only the FFN parameters are quantized.

| Quantization Setting | Prefix Length | Config | Prefix time (ms) | Decode per step time (ms) |
|---|---|---|---|---|
| Attention-only | 1024 | w16a16 | 299.958 | 26.178 |
| | | w8a16 | 302.199 | 25.710 |
| | | w4a16 | 301.899 | 25.595 |
| | 2048 | w16a16 | 616.460 | 26.532 |
| | | w8a16 | 616.080 | 26.117 |
| | | w4a16 | 615.601 | 25.964 |
| | 4096 | w16a16 | 1296.000 | 27.254 |
| | | w8a16 | 1290.000 | 26.685 |
| | | w4a16 | 1286.000 | 26.653 |
| FFN-only | 1024 | w16a16 | 299.958 | 26.178 |
| | | w8a16 | 292.630 | 17.410 |
| | | w4a16 | 284.680 | 15.635 |
| | 2048 | w16a16 | 616.460 | 26.532 |
| | | w8a16 | 604.941 | 17.868 |
| | | w4a16 | 602.306 | 16.006 |
| | 4096 | w16a16 | 1296.000 | 27.254 |
| | | w8a16 | 1276.000 | 18.521 |
| | | w4a16 | 1273.000 | 16.626 |

### F.3.2 RESULTS FOR ATTENTION + FFN QUANTIZATION

We now perform FFN + Attention weight quantization, results for which are presented in Table 18. We use the same experimental setting as the one described in Section 4. In our experiments we noticed that the inputs to the attention layer are often aligned in a handful of directions. Consequently, performing quantization using such a data leads to a huge drop in performance, as the algorithms would primarily focus on a few directions and ignore the rest. To mitigate this, we clip the largest eigenvalues of the Hessian to ensure a more balanced Hessian. This technique, reminiscent of the weight clipping in OmniQuant (Shao et al., 2023), improves the performance of both GPTQ and CDQuant[1]. For instance, for PaLM2-Gecko, the perplexity for w3a16-GPTQ improves from 34.872 to 24.054 with clipping, and for w4a16-GPTQ, it improves from 10.966 to 10.005. Table 18 presents the results from this experiment, where we run all the algorithms on the clipped Hessian. We once again notice that in almost all the settings, our coordinate descent algorithms outperform GPTQ. For instance, we see $5 - 10\%$ improvement in perplexity for 3-bit per-channel quantization of Gemma-2, and PaLM2 Gecko models.

---

[1]One could also rely on existing techniques such as AWQ and SmoothQuant to reduce the effect of outliers. But in our experiments we noticed that both AWQ and SmoothQuant performed poorly compared to the simple eigenvalue clipping technique.

Table 18: Table presents the perplexity evaluations for GPTQ, CD, BCD for INT3, INT4 quantization of both FFN and attention weights.

| Config | Method | Gemma-1 7B | Gemma-2 9B | Gemma-2 27B | PaLM2 Gecko | PaLM2 Otter | PaLM2 Bison |
|---|---|---|---|---|---|---|---|
| w16a16 | - | 10.348 | 10.683 | 8.682 | 7.948 | 5.984 | 5.298 |
| w3a16 | OWC | 1.89*e*6 | 12.558 | 15.736 | 41.751 | 54.499 | 6.446 |
| | GPTQ | 91.586 | 11.849 | 11.423 | 24.054 | **11.384** | 5.801 |
| | CD | 36.102 | 11.808 | 10.941 | 19.692 | 11.542 | 5.770 |
| | BCD(k=2) | **32.850** | **11.784** | **10.926** | **19.312** | 11.392 | **5.762** |
| w3a16g128 | OWC | 33.710 | 11.792 | 10.486 | 20.301 | 9.785 | 5.973 |
| | GPTQ | 20.045 | 11.491 | 9.766 | 15.042 | 7.501 | 5.698 |
| | CD | 15.850 | 11.500 | 9.689 | 14.757 | 7.454 | 5.679 |
| | BCD(k=2) | 15.466 | 11.487 | 9.657 | 14.691 | **7.438** | 5.673 |
| | OWC-CD | 18.599 | 11.583 | 9.820 | 16.268 | 7.617 | 5.746 |
| | OWC-CD + CD | 15.792 | 11.465 | **9.559** | 14.285 | 7.466 | 5.675 |
| | OWC-CD + BCD(k=2) | **15.777** | **11.463** | 9.564 | **14.242** | 7.458 | **5.670** |
| w4a16 | OWC | 40.563 | 11.100 | 9.706 | 10.683 | 7.194 | 5.508 |
| | GPTQ | 15.805 | 11.021 | 9.217 | 10.005 | 6.645 | 5.425 |
| | CD | 14.379 | **10.979** | 9.177 | 9.991 | 6.627 | 5.415 |
| | BCD(k=2) | **14.056** | 10.982 | **9.167** | **9.965** | **6.621** | **5.413** |
| w4a16g128 | OWC | 12.222 | 10.919 | 9.027 | 9.402 | 6.415 | 5.420 |
| | GPTQ | 11.548 | 10.863 | 8.888 | 9.160 | 6.241 | 5.381 |
| | CD | 11.306 | 10.856 | 8.890 | 9.124 | 6.226 | 5.377 |
| | BCD(k=2) | 11.105 | **10.853** | 8.892 | 9.120 | **6.225** | 5.376 |
| | OWC-CD | 11.342 | 10.876 | 8.924 | 9.241 | 6.261 | 5.387 |
| | OWC-CD + CD | 11.149 | 10.863 | 8.873 | 9.094 | 6.244 | 5.375 |
| | OWC-CD + BCD(k=2) | **11.140** | 10.866 | **8.873** | **9.093** | 6.242 | **5.374** |

## F.4 RUNTIME ANALYSIS

In Tables 19 20, we present the quantization runtime of GPTQ, CD and BCD on Gemma models using H100s (in one setting we used 1 H100 and in another we used 8H100s). In particular, in Table 19, we present runtime for quantizing all the attention and FFN layers (which includes FFN1, FFN1-gate, FFN2) in the model. Table 20 presents a more detailed breakdown; we present the per-layer runtime for quantizing FFN1 and FFN2 layers.

**Runtime on** 8 **H100's.** With 8 H100s, CD is as fast as GPTQ for quantizing attention weights, whereas BCD is $2\times$ slower. However, for FFN1 (FFN2) quantization, CD is $5\times$ ($2\times$) slower than GPTQ, whereas BCD is an order of magnitude slower than GPTQ. For quantization of the entire model, CD is $2\times$ slower than GPTQ (because quantizing FFN2 takes up most of the time). We attribute the slowness of CD and BCD to the numerous scatters and gathers involved in finding the most optimal coodrinate/block of coordinate to update, and then eventually updating it. Since ML accelerators are slow with gather and scatter operations, we are bound by the time it takes to do them. Performing these operations on the CPU should help speed up CD and BCD.

**Runtime on** 1 **H100.** With a single H100, GPTQ is 2x-3.8x faster than CD for the smaller, attention weights, and is an order of magnitude faster than BCD. For the larger, FFN weights, GPTQ is an order of magnitude faster than CD, and in most cases is two orders of magnitude than BCD. As stated previously, both CD and BCD are bound by the time it takes to do gathers and scatters.

**Speeding up BCD.** To algorithmically speed up BCD, we propose to use CD for FFN2 quantization, and use BCD for the rest of the layers. We find that BCD spends most of its time quantizing the FFN2 weight matrix (as can be seen in Table 20) where the quantization axis is of size $d_{\text{hidden}}$. In most cases, the time it takes to FFN2 is 10x the time it takes to quantize FFN1. Thus, to speed up BCD, we quantize FFN2 with CD, and FFN1 and FFN1-gate with BCD (where the quantization axis is of size $d_{\text{model}}$. Results for this setting can be found in Table 21. It can be seen that quantizing FFN2 with CD leads to negligible to no drop in performance especially for the larger, and better Gemma-2 9B. In fact, it improves the performance of BCD for w3a16g128. With this setup, BCD's runtime is reduced significantly and is comparable to that of CD.

**Speeding up CD.** We find that CD converges to an optimal solution significantly before $d_{in}$ iterations. Based on this finding, we run CD for lesser number of iterations. In Table 22, one can see that there is negligible quality drop for CD, and even with $d_{in}/8$ iterations, CD outperforms GPTQ. Also, one can see that in Figure 1, CD's activation reconstruction error converges in $d_{in}/8$ iterations across all settings. With lesser number of iterations, CD's runtime can be made faster than/comparable to that of GPTQ.

Table 19: Table presents the per-channel quantization runtime (in minutes) of GPTQ, CD, BCD (where each run is for $d_{in}$ iterations). FFN includes the combined runtime of quantizing FFN1, FFN1-gate and FFN2 layers.

| Config | Method | 8 H100s | | | | | | 1 H100 | | | | | |
| | | Gemma-1 7B | | Gemma-2 9B | | Gemma-2 27B | | Gemma-1 7B | | Gemma-2 9B | | Gemma-2 27B | |
| | | FFN (mins.) | Attn. (mins.) | FFN (mins.) | Attn. (mins.) | FFN (mins.) | Attn. (mins.) | FFN (mins.) | Attn. (mins.) | FFN (mins.) | Attn. (mins.) | FFN (mins.) | Attn. (mins.) |
|---|---|---|---|---|---|---|---|---|---|---|---|---|---|
| w3a16 | GPTQ | 1.68 | 0.67 | 1.15 | 1.05 | 7.98 | 1.18 | 2.13 | 0.65 | 1.36 | 1.04 | 10.31 | 1.32 |
| | CD | 3.03 | 0.19 | 2 | 0.31 | 16.55 | 0.51 | 22.3 | 1.2 | 13.31 | 1.87 | 122.35 | 3.2 |
| | BCD(k=2) | 8.52 | 0.71 | 5.95 | 1.18 | 66.97 | 2.11 | 88.14 | 3.85 | 61.5 | 7.24 | 541.03 | 13.62 |
| w4a16 | GPTQ | 1.69 | 0.77 | 1.15 | 1.06 | 7.95 | 1.18 | 2.1 | 0.63 | 1.36 | 0.98 | 10.16 | 1.12 |
| | CD | 3.9 | 0.22 | 2.53 | 0.36 | 21.16 | 0.64 | 29.27 | 1.47 | 17.66 | 2.25 | 163.78 | 4.27 |
| | BCD(k=2) | 26.2 | 1.76 | 18.51 | 2.66 | 162.43 | 4.64 | 218.25 | 12.89 | 163.64 | 20.12 | 1296.36 | 36.68 |

Table 20: Table presents the *per-layer*, per-channel quantization runtime (in seconds) of GPTQ, CD, BCD (where each run is for $d_{in}$ iterations) for the FFN layer. Note that FFN1 and FFN1-Gate have the exact same runtimes for all compared methods, hence we report only FFN1's runtime.

| Config | Method | 8 H100s | | | | | | 1 H100 | | | | | |
| | | Gemma-1 7B | | Gemma-2 9B | | Gemma-2 27B | | Gemma-1 7B | | Gemma-2 9B | | Gemma-2 27B | |
| | | FFN1 (sec.) | FFN2 (sec.) | FFN1 (sec.) | FFN2 (sec.) | FFN1 (sec.) | FFN2 (sec.) | FFN1 (sec.) | FFN2. (sec.) | FFN1 (sec.) | FFN2 (sec.) | FFN1 (sec.) | FFN2 (sec.) |
|---|---|---|---|---|---|---|---|---|---|---|---|---|---|
| w3a16 | GPTQ | 0.34 | 2.93 | 0.35 | 0.94 | 0.39 | 9.61 | 0.43 | 3.69 | 0.39 | 1.16 | 0.81 | 11.64 |
| | CD | 0.46 | 5.57 | 0.41 | 2.04 | 1.6 | 18.38 | 3.34 | 41.11 | 2.86 | 13.31 | 12.24 | 135.11 |
| | BCD(k=2) | 1.4 | 15.46 | 1.27 | 5.96 | 7.86 | 71.67 | 12.55 | 163.74 | 11.95 | 63.95 | 57.38 | 590.89 |
| w4a16 | GPTQ | 0.35 | 2.93 | 0.36 | 0.94 | 0.4 | 9.58 | 0.42 | 3.68 | 0.39 | 1.14 | 0.81 | 11.63 |
| | CD | 0.53 | 7.28 | 0.46 | 2.71 | 2.15 | 23.31 | 3.94 | 54.85 | 3.22 | 18.81 | 17.24 | 179.15 |
| | BCD(k=2) | 4.54 | 47.07 | 3.71 | 19.02 | 18.97 | 173.92 | 35.95 | 395.78 | 30.34 | 173.09 | 151.36 | 1388.19 |

Table 21: Table presents the perplexity evaluations for INT3 quantization with BCD wherein the FFN2 may be quantized either with CD or BCD, while rest of the FFN parameters are quantized with BCD.

| Config | Method | Gemma-1 7B | Gemma-2 9B |
|---|---|---|---|
| w16a16 | | 10.384 | 10.683 |
| w3a16 | CD, FFN2 | 19.709 | 11.288 |
| | BCD(k=2), FFN2 | 18.552 | 11.284 |
| w3a16g128 | CD, FFN2 | 13.512 | 11.179 |
| | BCD(k=2), FFN2 | 13.501 | 11.180 |

Table 22: Table presents the perplexity evaluations for INT3 quantization with CD, wherein instead of running CD for $d_{in}$ iterations, we run it for $d_{in}/2$, $d_{in}/4$, and $d_{in}/8$ iterations.

| Config | Method | Epochs | Gemma-1 7B | Gemma-2 9B |
|---|---|---|---|---|
| | w16a16 | | 10.384 | 10.683 |
| w3a16 | GPTQ | - | 48.157 | 11.382 |
| | CD | 1 | 19.614 | 11.301 |
| | | 1/2 | 19.614 | 11.301 |
| | | 1/4 | 19.621 | 11.301 |
| | | 1/8 | 19.597 | 11.305 |
| w3a16g128 | GPTQ | - | 15.561 | 11.193 |
| | CD | 1 | 13.496 | 11.182 |
| | | 1/2 | 13.488 | 11.189 |
| | | 1/4 | 13.480 | 11.189 |
| | | 1/8 | 13.560 | 11.188 |

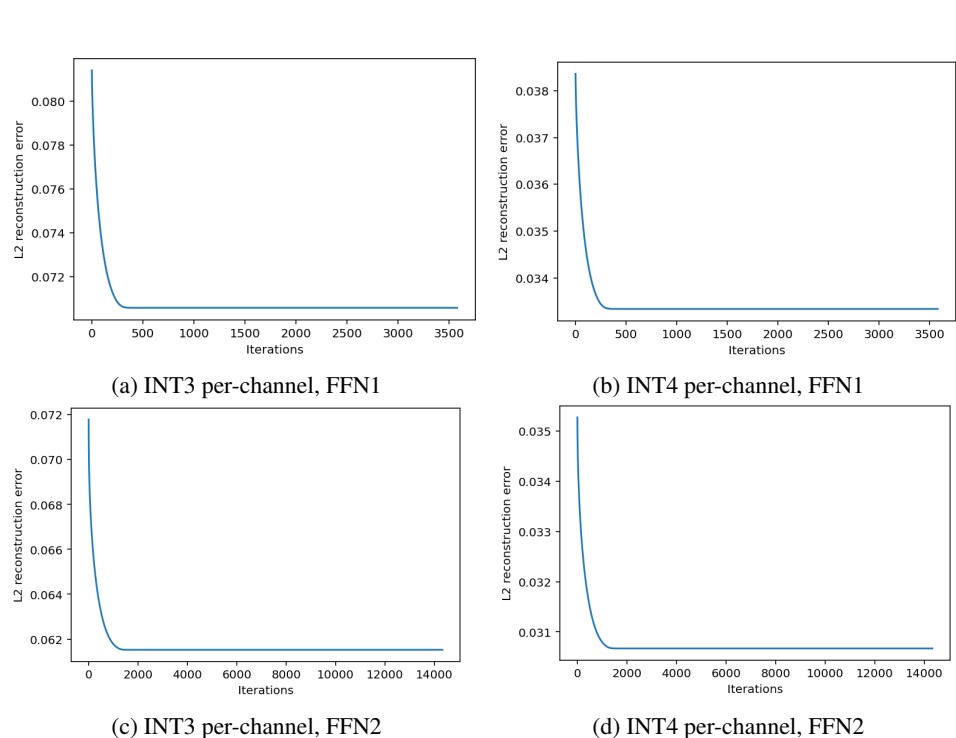

(a) INT3 per-channel, FFN1    (b) INT4 per-channel, FFN1

(c) INT3 per-channel, FFN2    (d) INT4 per-channel, FFN2

Figure 1: The Figure presents the variation of the L2 activation reconstruction error with iterations for for Gemma-2 9B with CD. One could see that across settings, CD converges in $d_{\text{in}}/8$ iterations

