# OpenReview forum: "CDQuant: Accurate Post-training Weight Quantization of LLMs using Greedy Coordinate Descent"
_ICLR.cc/2025/Conference — Submitted to ICLR 2025_

### Official Review · Reviewer_AoQn · 2024-10-17

**Soundness:** 3
**Presentation:** 4
**Contribution:** 2
**Rating:** 5
**Confidence:** 5

**Summary:**

The authors present CDQuant, an alternative to GPTQ that offers better performance in LLM quantization. CDQuant uses greedy coordinate descent to minimize layer-wise reconstruction loss, resulting in high-quality quantized weights. Evaluations on models such as Gemma and PaLM2 show that CDQuant outperforms GPTQ in 2-4 bit quantization. Furthermore, when integrated into other PTQ techniques like QuIP and FrameQuant, CDQuant enhances their performance as well.

**Strengths:**

1. This paper presents a set of improvements to GPTQ and introduces CDQuant, a straightforward and efficient alternative.

2. CDQuant demonstrates versatility and effectiveness when integrated with other PTQ techniques, such as AWQ, QuIP, and FrameQuant.

**Weaknesses:**

1. Actually, GPTQ has already identified the issue that OBQ updates coordinates in a greedy manner and introduced the Arbitrary Order Insight, which significantly reduces quantization time. Although CDQuant makes several efforts to accelerate quantization speed, it remains noticeably slower than GPTQ.

2. The experimental results suggest that, compared to GPTQ, CDQuant demonstrates certain advantages only on smaller models with relatively weaker capabilities, such as Gemma-1 7B and PaLM2-Gecko. However, it does not show clear advantages on models like Gemma-2, PaLM2-Otter & Bison, and it also lacks experiments on the LLama2 & 3 families, which are more commonly used in mainstream quantization research (e.g., GPTQ, QuIP, AWQ, etc.). Given that CDQuant uses more calibration data (1280 V.S. 128) and adopts OWC method, the benefits of using the greedy coordinate descent method remain uncertain.

**Questions:**

1. Since the LLama family models are more commonly used in LLM quantization research, could you provide CDQuant's results on LLama 2 and LLama 3 for a more intuitive comparison?

2. Currently, rotation-based quantization methods, such as QuaRot and SpinQuant, effectively eliminate outliers and demonstrate better performance than previous methods like AWQ and QuIP. Given that both QuaRot and SpinQuant use GPTQ for weight quantization, could you assess the versatility and effectiveness of CDQuant when integrated with these methods?

3. CDQuant uses 1280 samples for calibration, while 128 is more commonly used in other methods (e.g., GPTQ, AWQ, etc.). However, the authors did not explain this choice. Could it be because CDQuant leverages second-order information for error compensation and may overfit the calibration set during reconstruction?

---

> ### Author Response · Authors · 2024-11-17
> **Rebuttal**
>
> We thank the reviewer for their feedback. Below, we address some of the key concerns raised.
>
> **Comparisons to GPTQ:** As pointed out in our paper, the arbitrary order used by GPTQ usually leads to drop in quality, especially in extreme bit quantization. Our work tries to address this drawback of GPTQ via efficient greedy coordinate descent strategies. As pointed out by the reviewer, while OBQ also performs CD, its update rule is drastically different from ours and is very inefficient. This is because OBQ involves updating all the un-quantized coordinates in each iteration, which is very inefficient on accelerators such as GPUs, TPUs. As a result of this, the OBQ paper only managed to show experiments on ResNet models with a few million parameters. Our work addresses this drawback by providing efficient coordinate descent strategies which can scale to models with 100B parameters. Our algorithm is simple and only involves updating a single coordinate in each step.
>
> **Computational cost:** We would like to first note that on a multiple GPU setting, GPTQ and the CD variant of CDQuant have comparable runtimes. For FFN1 (FFN2) quantization, CD is 5x (2x) slower than GPTQ on 8 H100 GPUs. For the single GPU setting, the gap between their runtimes is more pronounced. However, as shown in Table 22 in Appendix F.4, CDQuant, even with 1/8th of the iterations, achieves better perplexity than GPTQ. Also, in Figure 1 in Appendix F.4 (please look at the updated paper), CD’s L2 activation reconstruction error converges in roughly 1/8th of the iterations across several settings. This reduction in iterations makes the runtime of CDQuant comparable to GPTQ. Moreover, in Table 21, in Appendix F.4, we show that replacing BCD with CD for FFN2 quantization does not lead to a significant performance drop. Since quantizing FFN2 is a bottleneck for BCD, this substitution significantly speeds it up. With the above two modifications, BCD can be considerably sped up, and CD could even run faster than GPTQ. As future work, we will work on writing kernels to implement the gathers and scatters in our algorithm efficiently.
>
> **Gains on model sizes:** We would like to highlight that our technique indeed shows significant gains on extreme quantization of large models (INT2 INT3 quantization). For instance, for INT2 quantization of PaLM2-Otter, we see a 10% improvement over GPTQ. Similarly, for INT2 quantization of Gemma2-27B and PaLM2–Bison, we see 3-5% gains.
>
> Furthermore, as we showed in our paper, CDQuant acts as an excellent replacement for GPTQ in algorithms that use GPTQ as a subroutine (see Table 3). For instance,  replacing GPTQ with CDQuant in QuIP, FrameQuant and AWQ showed 3-5% performance gains even on larger models such as Gemma2-27B. We would like to note that these results are highly non-trivial as (a) we are working with SOTA algorithms, and (b) providing a simple general purpose, plug-and-play approach to boost any SOTA technique that relies on GPTQ.
>
> **Experiments on Llama:** Unfortunately, due to certain organizational restrictions, we cannot use Llama models for research as that would violate the terms of use. To compensate for that, we run experiments with the PaLM2, Gemma-1 and Gemma-2 family of models. In  Riviere et al., 2024 [1], Gemma-2 27B has been shown to have comparable performance to Llama-3 70B. Furthermore, PaLM2 family models are production quality models which are usually very hard to compress. We thus believe PaLM2, Gemma-1 and Gemma-2 to be reasonable replacements for Llama-2 and Llama-2
>
> **OWC initialization:** We would like to highlight that, for a fair comparison, we used OWC initialization for both GPTQ and CDQuant. Thus, the performance gains observed in our experiments are due to the proposed coordinate descent algorithms.
>
> **Using 1280 data samples instead of 128:** We would like to note that our method works perfectly fine with 128 samples. We used 1280 samples for both GPTQ and CDQuant as it increased the Hessian approximation and improved the quantization performance for both CDQuant and GPTQ (this is especially the case with larger models with large FFN dimensions). CDQuant and GPTQ both leverage second-order statistics to quantize the LLMs and thus benefit from more examples. Furthermore, using 1280 data samples doesn’t add any significant computational overhead over using 128 samples and we do not see a strong reason for restricting ourselves to 128 samples. Ultimately, we care more about quality and thus went with 1280 samples in our work.
>
> [1] Riviere et al., 2024, Gemma 2: Improving Open Language Models at a Practical Size.

---

> > ### Author Response · Authors · 2024-11-17
> > **Rebuttal**
> >
> > **Comparison with recent PTQ techniques:** As we already demonstrated in our paper, CDQuant significantly outperforms GPTQ when used along with two other SOTA rotations algorithms, namely QuIP and FrameQuant. We expect the same behaviour to hold with both SpinQuant and QuaRot and other weight+activation quantization techniques as well. This is mainly because, once the appropriate rotation is applied to weights and activations, quantization of weights and activations is treated independently and performed using minmax or GPTQ or other popular techniques. We can simply replace these components with CDQuant for weight quantization.
> >
> > To demonstrate this, we provide W4A4 and W2A4 quantization results with QuaRot. Here, we replace the GPTQ component for weight quantization with CDQuant.  We observe that CDQuant consistently outperforms GPTQ. The results become more noticeable for 2-bits. We expect a similar trend to hold for SpinQuant as well.
> >
> > | Gemma-2 9B       |     | C4 perplexity  |
> > |------------------|-----|---------------|
> > | W16A16           |             | 10.683 |
> > | W16A4            |             | 11.01  |
> > | W2A4             | QuaRoT + GPTQ      | 13.571 |
> > |                  | QuaRoT + CD        | 13.311 |
> > |                  | QuaRoT + BCD       | 13.196 |
> > | W4A4             | QuaRoT + GPTQ      | 11.147 |
> > |                  | QuaRoT + CD        | 11.139 |
> > |                  | QuaRoT + BCD       | 11.122 |
> >
> > We hope we addressed the reviewer’s concerns, and are happy to answer any further questions. We would really appreciate it if the reviewer reevaluates our work.

---

> ### Comment · Reviewer_AoQn · 2024-11-26
>
> Thanks for the authors' response.
>
> Given the novelty of the method, which extends GPTQ by incorporating greedy search, and the limited improvement in accuracy, I will keep the original score.

---

> ### Comment · Reviewer_AoQn · 2024-11-27
>
> "But we are unsure why this should make our technique less novel compared to GPTQ."
>
> ---
>
> GPTQ, introduced two years ago, demonstrated the ability to quantize GPT models with 175 billion parameters in approximately four GPU hours, reducing the bitwidth of weights to as low as 3 or 4 bits with negligible accuracy degradation compared to the uncompressed baseline. This breakthrough significantly advanced the quantization accuracy and efficiency of LLMs, enabling—for the first time—the execution of a 175 billion-parameter model on a single GPU for generative inference.
>
> In comparison, I do not observe a comparable level of improvement and contribution from CDQuant.

---

> ### Comment · Reviewer_AoQn · 2024-11-27
>
> As shown in Table 7 and Table 8, under the configurations of w3a16g128 and w4a16g128, the accuracy of GPTQ, CD, and BCD is comparable.

---

> ### Author Response · Authors · 2024-11-28
>
> "As shown in Table 7 and Table 8, under the configurations of w3a16g128 and w4a16g128, the accuracy of GPTQ, CD, and BCD is comparable."
>
> --------------------------------------------------------------------------------------
>
> In Tables 7 and 8, the headroom for improvement was very little, because bf16 has similar performance as w4a16g128. For instance, for Gemma-2 27B, the avg bf16 accuracy is 67.55 and the avg accuracy with w4a16g128  is 67.45. So it's impossible to expect any technique to show improvements in such settings.
>
> That being said, for w3a16g128, we do show some improvements. For instance, for Gemma-2 27B, GPTQ has 66.4 avg accuracy, our technique has 66.87 and topline bf16 is 67.55. So, our technique bridges the gap between GPTQ and baseline. We believe, the results should be interpreted with topline in consideration.
>
> Finally, we would like to note that for settings where topline is significantly better than GPTQ, we do see significant gains.  For instance, for Gemma-1 7B quantization using w3a16, GPTQ has avg 40.87 accuracy, CD has 50.51, and topline bf16 has 58.35. Similarly,  for PaLM2-Gecko quantization using w3a16, GPTQ has avg 39.39 accuracy, CD has 41.12, and topline bf16 has 43.84 accuracy.
>
> Please also take a look at our INT2 quantization results where we have significant gains over GPTQ.

---

> ### Author Response · Authors · 2024-11-28
>
> We also request the reviewer to take a look at some of the recent papers on PTQ techniques.  For instance, Table 1 in AffineQuant paper (https://openreview.net/pdf?id=of2rhALq8l). The improvements over baselines diminishes as we increase the weight precision. This is expected because the gap between bf16 and baseline INT4 performance diminishes. The real value of these techniques shows up at extreme quantization (INT2, iNT3) and/or smaller models.
>
> Similarly, please look at Table 1 in the SqueezeLLM paper: https://arxiv.org/pdf/2306.07629, and Table 3 in MagR paper: https://arxiv.org/pdf/2406.00800.

---

> ### Comment · Reviewer_AoQn · 2024-12-01
>
> The authors highlight that CDQuant is intended as a replacement for GPTQ. While CDQuant demonstrates certain performance improvements over GPTQ in the w2 and w3 setting, the results have a considerable gap from floating-point accuracy and practical requirements.
>
> On the other hand, as they pointed out, under the w4a16g128 configuration (where CDQuant offers no advantage) , GPTQ achieves results very close to floating-point accuracy and satisfies the requirements of practical applications.
>
> I have always believed that there is room for improvement in GPTQ, and further advancements are necessary—such as addressing the model quantization generalization issues caused by the Hessian matrix. For this reason, I am always supportive of related research efforts.
>
> However, CDQuant provides limited theoretical innovation, particularly in terms of its integration of GPTQ and Greedy Search, and it falls short of fully addressing the core challenges associated with GPTQ. Moreover, the greedy search approach undermines its quantization efficiency. Overall, CDQuant fails to demonstrate sufficient potential to replace GPTQ.
>
> So the score (5) is appropriate, and I will maintain it.

---

### Official Review · Reviewer_9L8s · 2024-10-28

**Soundness:** 3
**Presentation:** 3
**Contribution:** 2
**Rating:** 5
**Confidence:** 5

**Summary:**

The paper presents CDQuant, a new quantization method for large language models that improves on GPTQ by using greedy coordinate descent to achieve better accuracy in low-bit quantization. It outperforms GPTQ in experiments on models like Gemma and PaLM2, providing lower perplexity and enhanced performance. CDQuant can also seamlessly replace GPTQ in other quantization techniques, making it a versatile and effective tool for compressing large models.

**Strengths:**

1. CDQuant consistently outperforms GPTQ in low-bit (2-4 bit) quantization, leading to better quantization quality and lower perplexity across various models.
2. The greedy coordinate descent approach in CDQuant provides better optimization of the layer-wise objective compared to GPTQ, leading to more efficient quantization.
3. CDQuant demonstrates significant performance improvements, especially on smaller models like PaLM2-Gecko and Gemma-1 7B, where it reduces perplexity by up to 10%.

**Weaknesses:**

1. CDQuant, particularly with Block Coordinate Descent (BCD), is significantly slower than GPTQ, especially for large models. The paper presents runtime comparisons (Table 5) showing that CDQuant is about 5× slower than GPTQ for FFN1 quantization and up to 10× slower for FFN2 quantization on models like Gemma-2 27B. BCD is even slower, with runtimes an order of magnitude higher than GPTQ in some cases.
2. For larger models, such as Gemma-2 27B, the computational cost of CDQuant becomes prohibitive. The time required to quantize the FFN2 layer, which has a larger quantization axis, is significantly higher than for other layers. This is demonstrated in Table 20, where FFN2 quantization takes up to 10× longer than FFN1 quantization.
3. Experiments based on a series of new models should be included in the paper. Would the llama series models, such as llama3, also be suitable for CDQuant?

**Questions:**

1. What are the theoretical guarantees of CDQuant's convergence? Does the greedy coordinate descent method guarantee convergence to a global minimum, or is it more prone to local minima, especially in high-dimensional spaces like those of LLMs?
2. How does CDQuant perform on models larger than those tested (e.g., models with trillions of parameters)? The paper demonstrates results on models with up to tens of billions of parameters (like Gemma-2 27B). How would CDQuant scale to models with trillions of parameters?
3. Why was MinMax quantization chosen as the baseline for comparison? The paper mentions that CDQuant uses Optimal Weight Clipping (OWC) for initialization, which performs better than MinMax quantization. Why not compare CDQuant's initialization with other advanced techniques like SmoothQuant or AWQ?
4. How does CDQuant perform when both weights and activations are quantized? The paper primarily focuses on weight quantization. How does CDQuant perform when both weights and activations are quantized, especially in scenarios like W4A4 quantization? How do the results compare with those of some recent quantization papers [1-3]?

[1] OmniQuant: Omnidirectionally Calibrated Quantization for Large Language Models. ICLR 2024.

[2] AffineQuant: Affine Transformation Quantization for Large Language Models. ICLR 2024.

[3] QuaRot: Outlier-Free 4-Bit Inference in Rotated LLMs. Arxiv 2024.

[4] SpinQuant: LLM quantization with learned rotations. Arxiv 2024.

---

> ### Author Response · Authors · 2024-11-17
> **Rebuttal**
>
> We thank the reviewer for their feedback. Below, we address some of the key concerns raised.
>
> **Computational cost:** We would like to first note that on a multiple GPU setting, GPTQ and the CD variant of CDQuant have comparable runtimes. For FFN1 (FFN2) quantization, CD is 5x (2x) slower than GPTQ on 8 H100 GPUs. For the single GPU setting, the gap between their runtimes is more pronounced. However, as shown in Table 22 in Appendix F.4, CDQuant, even with 1/8th of the iterations, achieves better perplexity than GPTQ. Also, in Figure 1 in Appendix F.4 (please look at the updated paper), CD’s L2 activation reconstruction error converges in roughly 1/8th of the iterations across several settings. This reduction in iterations makes the runtime of CDQuant comparable to GPTQ. Moreover, in Table 21, in Appendix F.4, we show that replacing BCD with CD for FFN2 quantization does not lead to a significant performance drop. Since quantizing FFN2 is a bottleneck for BCD, this substitution significantly speeds it up. With the above two modifications, BCD can be considerably sped up, and CD could even run faster than GPTQ. As future work, we will work on writing kernels to implement the gathers and scatters in our algorithm efficiently.
>
> **Experiments on Llama:** Unfortunately, due to certain organizational restrictions, we cannot use Llama models for research as that would violate the terms of use. To compensate for that, we run experiments with the PaLM2, Gemma-1 and Gemma-2 family of models. In  Riviere et al., 2024 [1], Gemma-2 27B has been shown to have comparable performance to Llama-3 70B. Furthermore, PaLM2 family models are production quality models which are usually very hard to compress. We thus believe PaLM2, Gemma-1 and Gemma-2 to be reasonable replacements for Llama-2 and Llama-2
>
> **Convergence guarantees:** We would like to note that the least squares minimization problem that we are solving at each layer is NP hard (see line 156 in our paper). This is a combinatorial optimization problem, and takes exponential (in dimension) time to converge to global optimum.  However, one can easily show that our algorithm converges to a special form of local optimum where modifying a single coordinate doesn’t reduce the loss value. The same can not be said of GPTQ which cycles through each coordinate only once. This is also the reason why we see an improvement in performance over GPTQ.
>
> **Performance on models with trillions of parameters:** Unfortunately, we do not have access to open source models with trillions of parameters or resources to run evals on them (getting perplexity evals and few shot evals requires a lot of resources). That being said, we believe similar performance trends observed in our paper will hold even for models with trillions of parameters. We base this claim on two main reasons: (1) SOTA trillion parameter models are MoE models, with each expert roughly the size of the largest model experimented in our paper, (2) the core problem we are solving (quantized least squares problem) remains the same across models. Consequently, we believe the gains we see here translate to bigger models.
>
> **Comparison between AWQ and OWC:** Thanks for the suggestion. Below we compare OWC and OWC-CD (the two initialization strategies introduced in our paper) and find them to outperform AWQ. The following table shows the perplexity numbers for Gemma-2 9B quantization using AWQ, OWC, OWC-CD. As can be seen, OWC, OWC-CD provide better initializations than AWQ (OWC-CD has blanks because it is designed only for sub-channel quantization).
>
> | Gemma-2 9B       | AWQ   | OWC   | OWC + CD |
> |------------------|-------|-------|----------|
> | W3A16            | 14.02 | 11.666| -        |
> | W3g128A16        | 11.449| 11.338| 11.22    |
> | W4A16            | 11.101| 10.929| -        |
> | W4g128A16        | 10.815| 10.82 | 10.786   |
>
> [1] Riviere et al., 2024, Gemma 2: Improving Open Language Models at a Practical Size.

---

> > ### Author Response · Authors · 2024-11-17
> > **Rebuttal**
> >
> > **Comparison with weight+activation quantization techniques:** As we already demonstrated in our paper, CDQuant significantly outperforms GPTQ when used along with two other SOTA rotations algorithms, namely QuIP and FrameQuant. We expect the same behaviour to hold with both SpinQuant and QuaRot and other weight+activation quantization techniques as well. This is mainly because, once the appropriate rotation is applied to weights and activations, quantization of weights and activations is treated independently and performed using minmax or GPTQ or other popular techniques. We can simply replace these components with CDQuant for weight quantization.
> >
> > To demonstrate this, we provide W4A4 and W2A4 quantization results with QuaRot. Here, we replace the GPTQ component for weight quantization with CDQuant.  We observe that CDQuant consistently outperforms GPTQ. The results become more noticeable for 2-bits. We expect a similar trend to hold for SpinQuant as well.
> >
> > | Gemma-2 9B       |     | C4 perplexity  |
> > |------------------|-----|---------------|
> > | W16A16           |             | 10.683 |
> > | W16A4            |             | 11.01  |
> > | W2A4             | QuaRoT + GPTQ      | 13.571 |
> > |                  | QuaRoT + CD        | 13.311 |
> > |                  | QuaRoT + BCD       | 13.196 |
> > | W4A4             | QuaRoT + GPTQ      | 11.147 |
> > |                  | QuaRoT + CD        | 11.139 |
> > |                  | QuaRoT + BCD       | 11.122 |
> >
> > **Comparison with other weight+activation quantization techniques:** CDQuant can be used to improve AffineQuant as it is again orthogonal to CDQuant. This technique uses Asymmetric MinMax quantization to quantize the rescaled weights. We expect CDQuant to work as a plug-and-play replacement for Asymmetric MinMax quantization and improve AffineQuant’s performance. Similarly, OmniQuant can be improved by replacing the Asymmetric MinMax quantizer with CDQuant.
> >
> > We hope we addressed the reviewer’s concerns, and are happy to answer any further questions. We would really appreciate it if the reviewer reevaluates our work.

---

### Official Review · Reviewer_8oR7 · 2024-11-03

**Soundness:** 3
**Presentation:** 3
**Contribution:** 2
**Rating:** 6
**Confidence:** 4

**Summary:**

The paper presents CDQuant, a quantization algorithm that improves the efficiency LLM through a greedy coordinate descent approach. CDQuant is an incremental work following GPTQ, a post-training quantization method by minimizing layer-wise reconstruction loss. The authors evaluate CDQuant across multiple model families, including Gemma and PaLM2, where it shows slight performance improvements over GPTQ in 2-4 bit weight-only quantization.

**Strengths:**

CDQuant proposes a unique greedy coordinate descent approach for LLM quantization, offering an alternative to the cyclic method used in GPTQ.

The experiments are comprehensive, covering multiple models and quantization settings.

The algorithmic description of CDQuant, as well as the variants, are clearly explained.

**Weaknesses:**

High computational cost: CDQuant’s runtime is much higher than GPTQ, especially on larger layers (e.g., FFN2). While the authors suggest mitigation strategies, further optimizations could enhance practicality.

Incremental Improvement: the idea is an incremental change to GPTQ, the improvement is also marginal compared with GPTQ, especially in the W4A16 setting which is the most practical use case.

**Questions:**

How does CDQuant perform in layers with extreme outliers compared to other approaches, like SqueezeLLM, which address outlier weights?

What modifications would be necessary to apply CDQuant to Quantization-Aware Training (QAT)?

---

> ### Author Response · Authors · 2024-11-17
> **Rebuttal**
>
> We thank the reviewer for their feedback. Below, we address some of the key concerns raised.
>
> **Computational cost:** We would like to first note that on a multiple GPU setting, GPTQ and the CD variant of CDQuant have comparable runtimes. For FFN1 (FFN2) quantization, CD is 5x (2x) slower than GPTQ on 8 H100 GPUs. For the single GPU setting, the gap between their runtimes is more pronounced. However, as shown in Table 22 in Appendix F.4, CDQuant, even with 1/8th of the iterations, achieves better perplexity than GPTQ. Also, in Figure 1 in Appendix F.4 (please look at the updated paper), CD’s L2 activation reconstruction error converges in roughly 1/8th of the iterations across several settings. This reduction in iterations makes the runtime of CDQuant comparable to GPTQ. Moreover, in Table 21, in Appendix F.4, we show that replacing BCD with CD for FFN2 quantization does not lead to a significant performance drop. Since quantizing FFN2 is a bottleneck for BCD, this substitution significantly speeds it up. With the above two modifications, BCD can be considerably sped up, and CD could even run faster than GPTQ. As future work, we will work on writing kernels to implement the gathers and scatters in our algorithm efficiently.
>
> **Improvements from CDQuant:** While we agree with the reviewer that improvements are less pronounced compared to GPTQ for W4A16, the gains are significant for extreme quantization (2 or 3 bit quantization; 5-10% gains in a number of settings). These extreme quantization settings have received significant attention of late and are crucial for widespread deployment of LLMs. We believe our technique can provide value in such settings. Additionally, as demonstrated in the paper, CDQuant can be used as a plug-and-play replacement for GPTQ in numerous state-of-the-art PTQ methods like QuIP, FrameQuant, QuaRot, and AWQ, and provides an easy way to boost their performance. We believe this last property makes our technique especially valuable in practice.
>
> **Using CDQuant on layers with extreme outliers:** In the presence of extreme outliers, we observed Hessian eigenvalue clipping helped us effectively handle the outliers (it helped boost GPTQ performance as well). The Hessian eigenvalue clipping is reminiscent of rescaling done in AWQ. This is also similar in spirit to methods like QuIP and FrameQuant that use rotation to handle extreme outliers. SqueezeLLM uses non-uniform quantization and a sparse-and-dense quantization scheme, both orthogonal to CDQuant. We expect CDQuant to benefit from these methods as well.
>
> **Applying CDQuant to QAT:** This is an interesting suggestion that we are planning to explore in the future. We believe coordinate descent based techniques could be a very good alternative to Straight Through Estimation (STE). There are various ways in which one could use CD for QAT. For instance, after each step of SGD (in full precision), one may take the full precision weights and use our technique to obtain quantized weights. On a related note, we would like to highlight that a recent NeurIPS 2024 oral paper by Malinovskii et al., [1] explores the use of greedy coordinate descent as an alternative to STE for vector quantization.
>
> We hope we addressed the reviewer’s concerns, and are happy to answer any further questions. We would really appreciate it if the reviewer reevaluates our work.
>
> [1] Malinovskii et al., 2024, PV-Tuning: Beyond Straight-Through Estimation for Extreme LLM Compression.

---

> > ### Comment · Reviewer_8oR7 · 2024-11-25
> >
> > the high computational cost is still concerning to me. I would keep the original score.

---

> > > ### Author Response · Authors · 2024-11-26
> > >
> > > We thank the reviewer for their prompt reply. As mentioned in our paper (as well as in the rebuttal), we provide two approaches to reduce the computational cost of CDQuant:
> > >
> > >  -  **running CD and BCD for fewer iterations**: As shown in Figure 1 in the updated draft, both these algorithms converge in very few iterations. Relying on this insight already gives us 8x reduction in computational cost without effecting the quality (please refer to table 22 for quality numbers). This makes CD as computationally efficient as GPTQ.
> > >  -  **replacing BCD with CD for FFN2 quantization**: BCD spends most of its time quantizing FFN2. To speedup BCD, one can rely on CD for quantizing FFN2. Please refer to Table 21 for quality numbers using this approach.
> > >
> > > The empirical results clearly show that both these approaches significantly speedup our algorithms without hurting the quality.  Please let us know, if you have any further concerns or questions or if the above arguments are not convincing.  We will be happy to allay your concerns.

---

### Meta-Review · Area_Chair_FkZL · 2024-12-23

**Metareview:**

It received mixed ratings of 6,5,5.

The reviewers pointed out several weaknesses including: CDQuant suffers from high computational cost, particularly on larger models like Gemma-2 27B, where its runtime can be up to 10× slower than GPTQ, especially for layers like FFN2. Despite efforts to optimize with Block Coordinate Descent (BCD), it remains significantly slower than GPTQ, even for smaller models. The improvements over GPTQ are marginal, particularly in the practical W4A16 setting. Additionally, CDQuant's advantage is mostly seen in smaller models with weaker capabilities, and it lacks experiments on more commonly used models like LLaMA 2 and 3, which limits its broader applicability. The reliance on more calibration data and the OWC method further raises questions about the efficiency and benefits of its approach compared to existing methods.

They also mentioned some positive points including the positive results compared to the baselines.

In the end, after the discussion, the reviewers still have the following concerns:
The main weaknesses of CDQuant lie in its limited theoretical innovation and insufficient improvements over GPTQ. While it incorporates a greedy search approach, this integration does not address key challenges in model quantization, such as the generalization issues caused by the Hessian matrix. Additionally, CDQuant's performance improvements are modest, and it fails to offer advantages over GPTQ in practical settings, particularly under the w4a16g128 configuration. The algorithm's high computational cost and slower efficiency further diminish its potential to replace GPTQ.

Since in the end, none of the reviewers is sufficiently positive about the paper, it will not be accepted.

**Additional Comments On Reviewer Discussion:**

All reviewers interacted in the discussion. We agree with the authors that the interaction was very limited and short for 2 of the reviewers, but unfortunately, we cannot ignore the negative points those reviewers have raised.

---

### Decision · Program_Chairs · 2025-01-22

Reject